

# Coastal Sea Level Monitoring in the Mediterranean and Black Seas

Begoña Pérez Gómez[1], Ivica Vilibić[2], Jadranka Šepić[3], Iva Međugorac[4], Matjaž Ličer[5], Laurent Testut[6], Claire Fraboul[7], Marta Marcos[8], Hassen Abdellaoui[9], Enrique Álvarez Fanjul[1], Darko Barbalić[10], Benjamín Casas[11], Antonio Castaño-Tierno[12], Srđan Čupić[13], Aldo Drago[14], María Angeles Fraile[15], Daniele A. Galliano[16], Adam Gauci[14], Branislav Gloginja[17], Víctor Martín Guijarro[15], Maja Jeromel[5], Marcos Larrad Revuelto[18], Ayah Lazar[19], Ibrahim Haktan Keskin[20], Igor Medvedev[21], Abdelkader Menassri[9], Mohamed Aïssa Meslem[9], Hrvoje Mihanović[22], Sara Morucci[23], Dragos Niculescu[24], José Manuel Quijano de Benito[18], Josep Pascual[25], Atanas Palazov[26], Marco Picone[23], Fabio Raicich[27], Mohamed Said[28], Jordi Salat[29], Erdinc Sezen[20], Mehmet Simav[20], Georgios Sylaios[30], Elena Tel[12], Joaquín Tintoré[11], Klodian Zaimi[31], George Zodiatis[32,33,34]

[1]Puertos del Estado, Madrid, Spain
[2]Ruđer Bošković Institute, Division for Marine and Environmental Research, Zagreb, Croatia
[3]Faculty of Science, University of Split, Split, Croatia
[4]University of Zagreb, Faculty of Science, Department of Geophysics, Zagreb, Croatia
[5]Slovenian Environment Agency, Ljubljana, Slovenia
[6]LIENSs, CNRS - La Rochelle University, La Rochelle, France
[7]SHOM, Brest, France
[8]Department of Physics, University of the Balearic Islands, Palma de Mallorca, Spain
[9]National Institute of Cartography and Remote Sensing, Algiers, Algeria
[10]Croatian Waters, Zagreb, Croatia
[11]SOCIB -Balearic Islands Coastal Ocean Observing and Forecasting System-, Palma, 07122, Spain. [12]Spanish Institute of Oceanography, Madrid, Spain
[13]Hydrographic Institute of the Republic of Croatia, Split, Croatia
[14]University of Malta, Department of Geosciences, Oceanography Malta Group, Msida, Malta
[15]National Geographic Institute, Madrid, Spain
[16]Joint Research Centre, Ispra, Italy
[17]Hydrometeorological and Seismological Service, Podgorica, Montenegro
[18]Spanish Hydrographic Office, Cádiz, Spain
[19]Israel Oceanographic and Limnological Research, Haifa, Israel
[20]General Directorate of Mapping, Department of Geodesy, Ankara, Türkiye
[21]PP Shirshov Institute of Oceanology, Moscow, Russia
[22]Institute of Oceanography and Fisheries, Split, Croatia
[23]ISPRA - Istituto Superiore per la Protezione e la Ricerca Ambientale, Roma, Italy
[24]National Institute for Marine Research and Development "Grigore Antipa", Constanța, Romania
[25]L'Estartit Meteorological Station, Gerona, Spain
[26]Institute of Oceanology, Bulgarian Academy of Sciences, Varna, Bulgaria
[27]CNR, Institute of Marine Sciences, Trieste, Italy
[28]National Institute of Oceanography and Fisheries, Alexandria, Egypt
[29]CSIC, Institute of Marine Sciences, Barcelona, Spain
[30]Democritus University of Thrace, Komotini, Greece
[31]National Centre for Forecast and Monitoring of Natural Risks, Polytechnic University of Tirana, Tirana, Albania
[32]ORION Research, Nicosia, Cyprus
[33]Institute of Applied and Computational Mathematics, Heraklion, Greece
[34]University of Cyprus, Cyprus Oceanography Centre, Nicosia, Cyprus



*Correspondence to*: Begoña Pérez Gómez (bego@puertos.es)

**Abstract.** Spanning over a century, a traditional way to monitor sea level variability by tide gauges is – in combination with modern observational techniques like satellite altimetry – an inevitable ingredient in sea level studies over the climate scales and in coastal seas. The development of the instrumentation, remote data acquisition, processing and archiving in last decades

allowed for extending the applications towards a variety of users and coastal hazard managers. The Mediterranean and Black seas are an example for such a transition – while having a long tradition for sea level observations with several records spanning over a century, the number of modern tide gauge stations are growing rapidly, with data available both in real-time and as a research product at different time resolutions. As no comprehensive survey of the tide gauge networks has been carried out recently in these basins, the aim of this paper is to map the existing coastal sea level monitoring infrastructures and the

respective data availability. The survey encompasses description of major monitoring networks in the Mediterranean and Black seas and their characteristics, including the type of sea level sensors, measuring resolutions, data availability and existence of ancillary measurements, altogether collecting information about 236 presently operational tide gauge stations. The availability of the Mediterranean and Black seas sea level data in the global and European sea level repositories has been also screened and classified following their sampling interval and level of quality-check, pointing to the necessity of harmonization of the

data available with different metadata and series at different repositories. Finally, an assessment of the networks' capabilities for their usage in different sea level applications has been done, with recommendations that might mitigate the bottlenecks and assure further development of the networks in a coordinated way, being that more necessary in the era of the human-induced climate changes and the sea level rise.

## 1 Introduction

Coastal sea levels have been monitored for decades by networks of tide gauges in ports and harbours, established by a diverse range of institutions to fulfil their specific needs and requirements. Tidal predictions, datum definition and port operations were the original motivation for creating most of these networks. However, tide gauge data are also needed to understand sea level changes at different spatial and temporal scales and are used by experts in fields such as oceanography, hydrography, meteorology, geodesy or seismology. The sea level sensors, which measure water level height relative to land with high

accuracy and high temporal resolution (1-60 min), are essential for monitoring and studying coastal sea level hazards that may threat the coastal strip during episodes of extreme sea levels and coastal flooding, the latter being combination of storm surges, tsunamis, meteotsunamis and infragravity waves occurring atop of ongoing sea level rise (Pugh and Woodworth, 2014). Improved knowledge and assessment of sea level changes in magnitude and frequency is essential for coastal planning, as well as for establishing early warning systems, for which tide gauges are a key element along with other *in situ* measurements and

forecasting models.



All mentioned hazards are present in the Mediterranean and the Black Sea (M/BS hereafter) and pose a threat to densely populated coastal areas, cultural heritage and historical cities lying near the shore (Fig. 1) (Reimann et al., 2018). This is particularly relevant for some regions exposed to substantial sea level variations spanning over a range of frequencies (Fig. 1), from minutes (like meteotsunamis or tsunamis) through hours, days, weeks (storm surges, or planetary wave forcing) and
seasonal oscillations, to interannual variability and decadal trends (Pugh and Woodworth, 2014).

As the Mediterranean and Black seas are microtidal basins (Tsimplis et al., 1995), the atmospherically-driven component of sea level (storm surge) is often the most common cause of extreme coastal sea levels. Conjoined with wind-generated waves and/or intense precipitation (also related to increased river discharge), flood risks during a storm surge may lead to devastating flooding events (Bevacqua et al., 2019). According to Cid et al. (2016), Tunisian (Gulf of Gabes), Aegean
and Adriatic coasts undergo the highest number of sea level extreme events per year. In the Black Sea, the western coast is the most exposed to storm surges (Bresson et al., 2018). Low elevation areas, deltas and sinking land areas (e.g. Venice Lagoon and Po Delta in Italy, Nile Delta in Egypt, Ebro Delta in Spain) are also subjected to the high flood risks during extreme events (Ferrarin et al., 2021; El-Fishawi, 1989; Hereher, 2015; Grases et al., 2020). Consequently, coastal zone management and protection bodies are carrying out extensive sea level measurements to support coastal flooding forecasts and issue timely
alerts to the population, like The Tide Monitoring and Forecast Centre of the City of Venice that maintains a network comprising tens of tide gauges. From time to time, exceptional high sea levels in the Adriatic Sea threaten particularly the city of Venice (*acqua alta* phenomena), causing severe flooding and disruption of people's lives. As an example, on November 12, 2019, sea level reached 1.89 m (~ 1.3 m surge contribution), the second highest storm surge since 1966 event (1.94 m) (Ferrarin et al., 2021). Occasionally, at other areas, less extreme storm surge events (~ 0.5 m) are also able to cause, in combination with
waves, substantial damage to infrastructure, coastal erosion and flooding episodes (e.g. the storm Gloria hitting the Spanish Mediterranean coast in January 2020: Amores et al., 2020; Pérez Gómez et al., 2021).

Tsunamis are amplified coastal sea level oscillations with periods ranging from minutes to hours, mainly generated by strong submarine earthquakes. The convergence of the African and Eurasian plates makes them a likely hazard in the Mediterranean (Tinti and Maramai, 1996; Tinti et al., 2001; Papadopoulos and Papageorgiou, 2012; Papadopoulos et al., 2014,
Maramai et al., 2014, 2019; Samaras et al., 2015), where around 10% of all tsunamis worldwide occur, being particularly destructive in the Hellenic Arc area (Fig. 1b). The earthquakes can reach a magnitude of 7.5 to 8 there, triggering > 5-6 m wave heights. The hazard is lower in the Western MS, but tsunami waves over 1 m can reach most of the M/BS locations (Sørensen et al., 2012; Álvarez-Gómez et al., 2011). A well-known ancient tsunami is the one generated by a strong earthquake (magnitude 8 - 8.5) off Crete in 365 A.D., that caused many deaths and damage in the Middle East and all the way up the
Adriatic Sea, and the destruction of Alexandria port and library (event t4 in Fig. 1b). Also, in Greece, around 1600 B.C. a giant tsunami triggered by the collapse of Santorini volcano is considered to have caused the end of the Minoan civilization in Crete (event t5 in Figure 1b). Tsunamis can also be generated by landslides and volcanic eruptions, but these events are mostly localised (e.g. Stromboli volcano, Tinti et al., 2005). Recent tsunami events (like the 9 July 1956 Amorgos 12-m high tsunami, Okal et al., 2009, the 21 May 2003 Algerian tsunami reaching the Balearic Islands shores as a 2-m wave, Alasset et al., 2006;



Vela et al., 2014, the 20 July 2017 Bodrum Peninsula and the 30 October 2020 Samos Island tsunami, Dogan et al., 2017, 2019) have been recorded by tide gauges. These records then became a valuable source of information for improving tsunami models and therefore for establishing and improving tsunami early warning services. As the tsunami consequences for the coastal population might be disastrous in terms of loss of life and economic damage, several regional tsunami service providers and national tsunami warning systems have been established in the area in recent years (e.g. Schindelé et al., 2015). These

systems, capable of assessing tsunamis travel time and providing early-warnings to civil-protection authorities on vulnerable coastal populations, make use of real-time tide gauge networks, for which 1 min or less sampling interval is required.

Meteorological tsunamis (also referred to as meteotsunamis) are atmospherically generated long-ocean waves which have spectral properties alike to those of tsunami waves, and which occasionally – at certain locations - reach destructive heights of tsunami waves (Monserrat et al., 2006; Rabinovich, 2020; Gusiakov, 2021). The Mediterranean Sea is considered

to be a meteotsunami hot-spot, i.e. a basin where a destructive meteotsunami occurs once in a decade or more often (Vilibić et al., 2021). The most researched Mediterranean meteotsunamis occur in Ciutadella harbour on Menorca Island (Jansà and Ramis, 2021), several locations along the eastern coast of the Adriatic Sea (Orlić, 2015; Orlić and Šepić, 2019), and at Mazara del Vallo on Sicily Island (Zemunik et al., 2021a). The Mediterranean meteotsunamis are commonly generated by high-frequency (T < 1 h) atmospheric gravity waves which, through the Proudman resonance (Proudman, 1929), generate long-

ocean waves while travelling over shelves. The strongest Mediterranean events typically occur during summer months (Vilibić et al., 2021) when synoptic situations which favours generation and propagation of atmospheric gravity waves are more common. Several attempts to construct a reliable meteotsunami warning system for the Mediterranean have been made, all for the Balearic Islands and the Adriatic Sea: these warning systems are designed to have at least one of the listed: (i) real-time monitoring of the atmosphere-ocean conditions (Marcos et al., 2009a; Šepić and Vilibić, 2011); (ii) numerical modelling of

atmospheric and ocean processes (Denamiel et al., 2019a; Romero et al., 2019), (iii) assessment of forecasted synoptic conditions (Jansà et al., 2007; Šepić et al., 2016b), or (iv) combination of some of the above (Denamiel et al., 2019b).

Sea level rise, a key indicator of ongoing climate changes, is an underlying threat to some coastal areas in the M/BS, particularly to those already exposed to extreme events and low elevation areas (especially if accompanied by subsiding land). Despite the small number of stations with sufficiently long time series for climate studies, historical monthly mean sea levels

from tide gauges have been used to compute coastal sea level trends (Zerbini et al., 1996, Tsimplis and Spencer, 1997; Gomis et al., 2012). These data provide relative sea level trends, relevant for assessing flooding risk and improving coastal protection, often differ significantly from one place to another due to local land movements (El-Geziry and Said, 2020). Absolute sea level trends can be obtained for those tide gauge stations that are co-located with permanent Global Navigation Satellite System (GNSS) stations that provide the vertical land motion correction (VLM). Tsimplis et al. (2005) found tide gauge sea level

trends of just -0.4 to 0.7 mm/yr between 1958 and 2001 in the Mediterranean, revealing slower sea level rise than is the global average for the period. This appears to be due to a negative trend of the atmospheric (storm surge) component, which in turn was caused by prolonged positive phase of the North Atlantic Oscillation (NAO). However, the Mediterranean sea level trends have increased significantly since the 1990s. Bonaduce et al. (2016) found for the Mediterranean basin a mean sea level positive



trend of 2.44 +- 0.5 mm/yr, based on satellite altimetry and tide gauge data for the period 1993-2012. Taibi and Haddad (2019)
used 18 tide gauges in the region with data spanning the period of 1993-2015 and found significant trends ranging from 1.48
to 8.72 mm/yr, after VLM correction, pointing to large spatial differences in the sea level rise.

The majority of the quoted research relies strongly on the tide gauge data, either directly or through their usage in
calibration of satellite altimeters. Near-real time data transmission, combined with a progressive upgrade to shorter temporal
sampling step, have allowed over the last 15 years the integration of tide gauge data in storm surge and tsunami warning
systems. Tide gauge data are also required for validation of global, regional and coastal circulation and tsunami models, coastal
engineering or altimetry data calibration. Their multi-purpose and multidisciplinary character are an advantage for the
sustainability of the system, ensuring a rather permanent funding in some cases. However, it also presents challenges for
basin/regional/global scale network coordination initiatives, and for data exchange between existing international programs.
This is particularly the case in the M/BS, where in some countries restrictive national data policies, especially along the
Mediterranean coast of Africa, have yielded a spatial distribution of stations with available data biased towards the northern
countries (Tsimplis and Spencer, 1997, Woodworth et al., 2009). Several attempts and efforts  in the past tried to solve this
situation: in 1997, the International Commission for the Scientific Exploration of the Mediterranean Sea (CIESM) and the
Intergovernmental Oceanographic Commission from UNESCO (IOC/UNESCO) agreed to cooperate in the research on sea
level changes in the M/BS by establishing a long-term monitoring network system named MedGLOSS, connected to the Global
Sea Level Observing System (GLOSS) (Rosen and Aarup, 2002, http://www.ciesm.org/marine/programs/medgloss.htm). The
programme, very active between 2001 and 2005, fostered the establishment of a network in the region, supporting data
contribution to international programs, digitization of old chart records, support on quality control, and even installation and
calibration of stations in some countries. Despite the fact that the number of stations has increased significantly in recent years,
and that the networks in many countries have been modernised  (with modernisation mainly driven by new requirements of
tsunami warning systems), there is still an important lack of available data along the southern coast of the basin. Other more
recent initiatives launched in the framework of MONGOOS or the IOC/UNESCO have also failed, up to now, to fill this gap.
Some national sea level networks were substantially upgraded recently following the technological development (e.g. the
Spanish networks: Pérez Gómez et al., 2013; 2014, Italian,French and Croatian radar networks, etc.), providing the data to
users following FAIR (Findable, Accessible, Interoperable and Reusable) principles recently established as the standard in the
science (Wilkinson et al., 2016).

Acknowledging all these developments, it emerged that a cohesive mapping of *in situ* coastal sea level monitoring
capacities in the M/BS basins has not been done for a long time, and even sporadically on national levels (e.g. Vilibić et al.,
2005), exceeding the time scale of the technological developments that are rapidly changing the observational landscape in
geosciences in general (Le Traon, 2013). The aim of this paper is to assess coastal sea level monitoring capacities at national
and basin-scale levels, to survey the availability of the sea level data, to address the appropriateness of the networks for the
most relevant sea level applications, and to identify required upgrades, maintenance problems and potential fields of regional
cooperation. To achieve this, a survey of coastal sea level infrastructure was conducted in 2021, including most of the relevant



institutions operating tide gauges in the region. Results of the survey are presented, including information on type of technology, ancillary measurements, co-location with GNSS, data sampling and latency, long term data availability, quality

control and funding status. The initiative enabled us to access relevant metadata and reach national contacts, and to improve communication and exchange of experiences between national experts in sea level studies and tide gauge operators. The survey has been complemented with an assessment on data availability in different international data portals and programs targeting different applications, and an analysis of the fit-for-purpose status of the network based on data availability at those areas more threatened by storm surges, tsunamis, meteotsunamis and sea level rise. Following introductory section, Section 2 comprises

a description of the tide gauge networks or stations operated by the different contributors and presents the main results of the survey. Section 3 is dedicated to data availability in existing international programs, databases and data repositories. Section 4 provides an assessment of the network to fulfil targeted applications, while Section 5 summarizes the work with several conclusions and recommendations.







**Figure 1. (a)** Maximum measured sea levels heights (with respect to a local mean), based on GESLA Version 3 dataset and available national databases, are given (coloured circles); strips of coasts (red lines) and the UNESCO heritage sites (white circles) endangered by present and future-day sea level rise and erosion (after Reimann et al., 2018); **(b)** Epicentres of earthquakes which resulted with historic tsunamis having intensities of 3 or higher (coloured circles, after Maramai et al., 2014); tsunamigenic fault areas are depicted with white rectangles, and tsunamigenic potential indicated with numbers 1-4 (after Fokaefs and Papadopoulos, 2007, and Oaie et al., 2016); locations of historic meteotsunamis surpassing amplitude of 0.5 m are marked with coloured stars (after Vilibić et al., 2010; 2016; Orlić, 2015; Šepić et al., 2018; Okal, 2021). For extreme hourly sea levels and meteotsunamis, locations at which most extreme events occurred are named; for tsunamis, epicentres of earthquakes leading to the eight most destructive tsunamis are marked with t1-t8.



## 2 National tide gauge networks and observations

Operating institutions and operators of tide gauges networks and observing sites in the M/BS region have been identified and contacted throughout 2021. We have received input from 30 different institutions, resulting with brief network descriptions, list of station details (Appendix) and additional questionnaire responses (Supplementary material).

### 2.1 Networks description

#### 2.1.1 Ports of Spain REDMAR network

The REDMAR network, composed today of 41 multi-purpose radar stations, was established by the Spanish harbour authorities and Puertos del Estado (PdE) in 1992, as an aid to port operations and coastal and harbour engineers. The first three stations in the Mediterranean coast were installed that year at Barcelona, Valencia and Málaga, based on acoustic sensors. Two new stations based on pressure sensors were deployed at Ibiza and Motril in 2003 and 2004, respectively. Ibiza was the first one co-located with a GNSS permanent station, in the framework of ESEAS-RI project, for altimetry calibration (Martínez-Benjamín et al., 2004).

Today, 17 stations are operated along the Spanish Mediterranean coast, including one at Melilla (North of Africa) and 5 at the Balearic Islands. Between 2005 and 2009 all the old stations were upgraded to radar sensors (and 1 min sampling and latency), after an overlapping period of around one year to connect and refer to the same datum the old and the new timeseries (Pérez Gómez et al., 2014). Ancillary atmospheric pressure and wind data (1 min resolution) are measured at 12 of the 17 stations and 7 of the stations are today collocated with a permanent GNSS receiver: Barcelona, Tarragona, Ibiza, Mallorca,Almería, Melilla and Tarifa. Data is displayed through the PdE visualization tool (Portus: https://portus.puertos.es).

All the stations transmit 1 min data with 1 min latency to PdE, the National Geographic Institute (National Tsunami Warning System) and the IOC Sea Level Station Monitoring Facility (SLSMF). Automatic quality control and processing is applied every 15 min for integration in the multi-model sea level forecasting system (Pérez Gómez et al., 2021). In addition, 2 Hz raw data are processed every hour to characterize higher frequency sea level oscillations with periods of minutes (García-Valdecasas et al., 2021). Delayed-mode quality-control and processing is performed annually and monthly mean sea levels are sent to the Permanent Service for Mean Sea Level (PSMSL). The data are also available through CMEMS INS TAC (NRT product), EMODnet and GESLA datasets. Mediterranean stations operated by PdE are listed in Table A1.

#### 2.1.2 SOCIB tide gauge network in the Balearic Islands

The SOCIB tide gauge network (Tintoré et al., 2013; 2019) was established in 2009 and currently consists of six stations around the Balearic Islands, five of which are located on Mallorca and one on Ibiza. The length of the nearly-continuous tide gauge records varies between 5 and 12 years, with the shortest series dating back to 2016. With the exception of the tide gauge in Sant Antoni (Ibiza) that is a radar gauge, the rest are pressure gauges. The sampling frequency is 1 min for all tide gauges. Together with sea level, all stations measure atmospheric pressure, sampled every 30 s, and five of them also monitor water





temperature every minute. All data are freely distributed in near-real time through the SOCIB website and are also available
through the CMEMS INS TAC data portal. Sea level observations are referenced to the tide gauge benchmarks whose positions
are controlled on a yearly basis through GNSS surveys. Stations operated by SOCIB are listed in Table A2.

### 2.1.3 Spanish National Geographic Institute tide gauge network

The tide gauge network of the National Geographic Institute of Spain (IGN) has a set of sensors that gather changes and
variations of the mean sea level over the time. It started in the 19th century, when three tide gauges were set up in order to
determine the national altimetry datum in Alicante, Santander and Cadiz. The purpose was to establish the required
infrastructure to start levelling works. The tide gauge network has been extended and its instruments have been improved since
then, including recent upgrades from float to radar sensors. Nowadays the National Geographic Institute has ten tide gauges:
five are located in Iberian Peninsula, one in Alboran Island and four in the Canary Islands. Only five are on the Spanish
Mediterranean coast. All of them have one or two radar sensors and are linked to GNSS permanent stations. A sea level dataset
starting in 1870 has been recently published for Alicante, based on data from historic float gauges and modern radar sensors
at this harbour (Marcos et al., 2021). The Mediterranean part of the network is complemented by five IDSL in Cartagena, La
Mola de Mahon, Ciutadella, Ceuta.

In addition to maintenance works, network management and connection to High Precision Levelling Network
(REDNAP), National Geographic Institute is analysing the historical series of its tide gauges. It is also comparing mean sea
level changes with GNSS observations among which a new technique is standing out. It is called GNSS reflectometry (GNSS-
R) and it is still under study. Mediterranean stations operated by IGN are listed in Table A3.

### 2.1.4 Spanish Institute of Oceanography tide gauge network

The sea level data network operated by the Instituto Español de Oceanografía (IEO) consists of 11 stations, six on the Iberian
Peninsula, one at Ceuta on the Northern Africa coast, one in the Balearic Islands and 3 in the Canary Archipelago. For historical
operative reasons, most of these locations are those in which IEO local headquarters are located. Each tide gauge station is
equipped with two sensors: an analogue float-type tide gauge with digital encoder and a radar-based one.

The analogue network is one of the oldest ones in Spain, with some of the measurements dating back to 1943.
Historical data are made available through SeaDataNet data portal (www.seadatanet.org). Each of the analogue sensors consist
of a float gauge, mechanically connected to an analogue-digital encoder which converts the data to an on-site computer. The
radar sensors duplicate the measurements and ensures the measurements continuity when one of the two devices fails. 1 min
sampled data are locally stored and recovered via modem by the central data centre once a day, where the data quality is
assessed and archived.

The routines for data recovery, quality assessment, detection of high frequency events and data representation are
currently being updated, and it is expected that the frequency of data availability will reach 1 per minute by the end of the year
2021. Mediterranean stations operated by IEO are listed in Table A4.



### 2.1.5 Spanish Hydrographic Office

Currently, the Spanish Hydrographic Office (IHM) is making a great effort in the installation of a permanent tide gauge network along the national coast. Usually, the IHM installs tide gauges in the hydrographic works area on a temporary basis to be able to calculate the real or reduced probe data. Once the work is finished, the tide gauge is usually removed.

270        Since 2021, the IHM has installed permanent tide gauges in different ports on the coast: Rosas, Castellón (in the Mediterranean coast) and Huelva (in the Gulf of Cadiz). The intention is to install 8 new tide gauges during 2022, with the objective to further increase density of tide gauges along the Spanish coast. The equipment consists of acoustic sensors with a frequency of 1 Hz and with spatial positioning in real time through a GNSS. At the moment, the web is under development. Mediterranean stations operated by IHM are listed in Table A5.

### 275  2.1.6 L'Estartit tide gauge (Meteolestartit, in collaboration with ICM/CSIC-Spain)

The tide gauge is a part of the Meteorological and Oceanographic Station (Meteolestartit) at the harbour of L'Estartit, a coastal town at the Catalan coast, in the NW Mediterranean. It is a float gauge with an analogical record on paper. Recordings are collected every week and digitised with a 2 h resolution. Paper records are preserved for further detailed analyses, if required in some special circumstances, such as seiches. The position of the tide gauge is georeferenced every 5 years by the Catalan
Cartographic Institute and sea level data is backwards linearly corrected for each period. Sea level record collection at this point started in January 1990, as part of Meteolestartit, which started in 1969, as a personal initiative of Josep Pascual with the collaboration with the Marine Sciences Research Institute in Barcelona (ICM/CSIC). Data collected included basic meteorological and oceanographic data. More details can be found in Pascual and Salat (2019) and Salat et al. (2019). Details about L'Estartit tide gauge are in Table A6.

### 285  2.1.7 SHOM tide gauges network RONIM, France

French Naval Hydrographic and Oceanographic Service (SHOM) has been observing the tides for many years, but the RONIM network, as it exists today, was initiated in 1992, mainly to meet the needs of tide prediction and reduction of bathymetric surveys. It was based on a network which was already in place and which consisted, in 1996, of five tide gauges. It was then densified to reach 23 tide gauges in 2007, including 5 in the Mediterranean (Marseille, Toulon, Nice, Monaco and Ajaccio).
Some tide gauges were installed on sites where SHOM already had sea level observations, allowing for continuation of measurements and prolongation of time series. At that time, the Mediterranean tide gauges were equipped with acoustic sensors.

       Today, the RONIM network consists of 50 stations of which 14 are currently active in the Mediterranean Sea. All tide gauges are equipped with radar sensors and also have an atmospheric pressure sensor. In the Mediterranean, 3 tide gauges
are collocated with a GNSS (Sète, Marseille and Toulon). Most of the tide gauges are equipped with a double real time transmission system: internet and satellite. In the Mediterranean, all tide gauges except Marseille are equipped with satellite



transmission. The acquisition rate is 1 Hz, and these data are sent directly to SHOM and tsunami warning service by a VPN link.

The 1 min data are computed at SHOM and transmitted in near-real-time to the data.shom.fr website and to the IOC
website with a latency of 5 min. The satellite messages are clocked every 6 min. The data transmitted by satellite are also on the IOC website. As there is a double transmission system, on the IOC site, the stations are in double (ex: Toulon (internet transmission) and Toulon2 (satellite transmission)).

The 10 min data is calculated by the tide gauge data logger. They are retrieved once a day and transmitted to the data.shom.fr website.

The RONIM network data are also available on CMEMS INS TAC, EMODNET, GESLA, SONEL, PSMSL data portals.

A major upgrade of the RONIM network is underway (2021/2022). The data logger and the real time transmission process will be replaced, and two additional sensors (webcam + meteo station) will be added at most of tide gauges. A unique supervision system will be implemented for all tide gauges, allowing for a better assessment of real time recording and
transmission issues and an improvement of the network reliability. Mediterranean stations operated by SHOM are listed in Table A7.

### 2.1.8 ISPRA tide gauge networks along the Italian coast

The Italian Institute for Environmental Protection and Research (ISPRA) comprehensively and systematically provides high-resolution estimates for the physical state of the Italian seas as well as real-time monitoring at national and local level. The
marine observation system includes two sea level measurement networks: Italian Tide Gauge Network (Rete Mareografica Nazionale - RMN), which continuously monitors the sea level and a number of related meteorological and physical parameters, and the North Adriatic and Venice Lagoon Tide Gauge Network (Rete Mareografica della Laguna di Venezia e del Litorale Nord Adriatico - RMLV) which is used for the real-time storm surge prediction and warning system.

The RMN network is a crucial source of information related to sea level. It provides data useful for analysing sea
level variations, predicting storm surges and developing a tsunami early warning system. The RMN consists of 36 measuring stations uniformly distributed along the Italian coast, mainly located within harbours. Some of these stations have been operating since the 1970s. The tidal-wave measurements for the entire network are provided by two different instruments (radar and float) and can be simply configured by a remote command. Each measurement station is also equipped with meteorological sensors: anemometer (wind speed and direction at 10 m above ground level), barometric sensor,
multiparametric sensor for humidity and temperature, that are necessary for the real-time evaluation of sea and weather conditions. Four stations (Venezia, Crotone, Gaeta, Carloforte) are collocated with GNSS instruments, to detect the horizontal and vertical displacement of the cabin.

The RMLV network is composed of 26 tide gauge stations equipped for the systematic and widespread measurement of water level and other related parameters, such as wind direction, wind speed, atmospheric pressure, precipitation, and wave-





heights in the Lagoon of Venice and along the North Adriatic coastline. Lots of the RMLV stations have been operating for several decades and Venezia - Punta della Salute station for more than 150 years. Real-time data represent one of the main utilities, fundamental for prediction and warnings of exceptional or atypical high tides (storm surges). Two RMLV stations (Venezia Punta della Salute and Grado) are collocated with GNSS in order to detect the continuous vertical shift of the local Zero Tide Level which is the reference benchmark for tide measurements in the Lagoon of Venice (ZMPS). The real-time

operability of this network is crucial for several purposes such as: analysis and elaboration of data referring in particular to extreme events (storm surges), signalling and forecasting exceptional high tides. Moreover, data from the Venice Lagoon and North Adriatic tide gauge network (RMLV) is an important source for planning Venice defence from the phenomena of high tides and for scientific studies on sea level variations.

Some IDSL stations were also tested, but the only one still operational is in Marina di Teulada.

Quality-control procedures have been implemented in order to validate sea level historical series and to guarantee data compliance with the international standards. Furthermore, automatic quality control procedures are applied on real-time data. Data are distributed through ISPRA portals (http://dati.isprambiente.it, www.mareografico.it, www.venezia.isprambiente.it and https://tsunami.isprambiente.it/, that is powered by the JRC TAD server software) and through international initiatives (IOC, EMODnet, MONGOOS, EuroGOOS). Stations operated by ISPRA are listed in Table

A8.

### 2.1.9 The Joint Research Center (JRC) IDSL network

Acknowledging the low quantity of sensors deployed on the Mediterranean coasts, the JRC started investigating the adoption of retail technology in 2014, in order to produce a low-cost solution. Since 2014, with the adoption of a few custom components, the reliability of The Inexpensive Device for Sea Level (IDSL) measurements further evolved retaining its best

characteristics: low cost (about 2k€), high frequency (5 secs), local intelligence (detection of anomalous sea oscillations through assessing deviation from moving averages).

Funded by the IOC-UNESCO, three campaigns delivered devices throughout the Mediterranean, the Black Sea and the North-East Atlantic. Another collaboration led to providing Indonesia with six IDSL devices. All data collected by the IDSL network are available online through the TAD Server at the JRC web site. Stations operated in Italy by JRC with ISPRA

are listed in Table A9.

### 2.1.10 The tide gauge station of Trieste, Molo Sartorio, Italy

The Italian National Research Council (CNR) operates one station through the Trieste branch of the Institute of Marine Sciences (ISMAR). The station, included in the GLOSS Core Network with No. 340, is located at Molo (Pier) Sartorio, in the harbour of Trieste, and it is equipped with two float instruments with digital encoder, and one 50-year old fully analogue tide

gauge. Direct sea level measurements are performed at least twice a month to check the instrument stability. The data quality control is performed in delayed mode at least once a year.



The tide gauge cabin also hosts two barometers, one of which is digital and one analogue. A GNSS receiver, operated by the University of Bologna, is mounted on top of the building that includes the tide gauge cabin.

The earliest sea level measurements were made in 1859, and since then the tide gauge remained on the same pier;
more details can be found in Zerbini et al. (2017). Unfortunately, from 1875 to 1939, only monthly mean sea levels are available (with a few gaps); they can be retrieved from the Permanent Service for Mean Sea Level (PSMSL) data bank. Hourly data are available for 1939-onwards (Raicich, 2019), while 1 min data exist since 2001. Hourly means are available as Fast Delivery data from the University of Hawaii Sea Level Center (UHSLC) since June 2009. The review of pre-1939 data including the digitisation of the available charts is in progress. Details about station Trieste Molo Sartorio are in Table A10.

**2.1.11 The mareographic station in Koper, Slovenia**

Operational sea level monitoring in Slovenia began in 1958 with the construction of the tide gauge station in Koper, for which hourly sea level measurements are available since 1961. The station is operated by the Slovenian Environment Agency and is collocated with the GNSS station. The existing float-type sensor in a stilling well was upgraded in 2005 with an additional radar sea level sensor, having 1 mm accuracy and 10 min sampling time. It provides sea level data, sea temperature data at 1
m of depth, GNSS data and essential meteorological data (air pressure, wind, air temperature, relative humidity, solar irradiance). The quality-control is automatic, while additional manual controlling of sea level measurements is performed weekly at the station location. Details about station Koper are in Table A11.

**2.1.12 Croatian tide gauge network**

The Croatian tide gauge network is operated by the Andrija Mohorovičić Geophysical Institute (AMGI), Hydrographic
Institute of the Republic of Croatia (HHI), Institute of Oceanography and Fisheries (IOF), National Agency for Water Management Hrvatske vode (Croatian Waters, HV), and by the Ruđer Bošković Institute (RBI). The network consists of permanent stations based on float-type technology installed in stilling wells, established during the 20th century, and of radar-type stations installed from 2004 onward.

*Float-type stations:* Systematic measurements of sea level in Croatia started in 1929 when the AMGI (Zagreb)
established a tide gauge station in Bakar. In the same year, sea level measurements were initiated in Split Harbor by the HHI, but the station was destroyed by bombing in World War II. During the 1950s four long-term coastal stations were installed: Dubrovnik (1955), Split Harbour (1955) and Rovinj (1955) by the HHI, and Split Marjan (1956) by the IOF. The network was again extended when the HHI installed station Zadar in 1990 and Ploče in 2002, and HV installed stations near Mala Neretva river mouth in 1977, at Prosika in 1986 and at Golubinka in 1995. Most of the network was modernized in the early 2000s by
mounting a/d (analogue/digital) converters to floats, along with GSM modems for real/near-real-time data acquisition. Modernization was mostly done in the frame of ESEAS-RI project (EU Framework Programme 5). In 2004 an open-air radar sensor was installed at Bakar station and GNSS at Split Harbor station. Ancillary atmospheric pressure (Bakar and Split Marjan) and wind (Split Marjan) measurements are carried out too. Most of the stations locally store 1 min data and transmit





them with up to 15 min latency to home institutions. Exceptions are stations Golubinka and Prosika from which data is collected
on-site at least once a year. At most stations digital data is automatically quality checked, processed, and displayed
(http://geo101.gfz.hr/~bakar/index_files/, https://adriaticsea.hhi.hr, http://vodostaji.voda.hr/) in real-time mode or once a day
(Bakar), while analogue data are digitized at hourly resolution, and mostly processed once a year. Maintenance of stations
includes regular checks of recorder zero (mainly twice a year), cleaning of the stilling-well and connecting pipe (once in two
to five years) and levelling of the contact-point level against tide gauge benchmarks. All stations are levelled towards the
national geodetic datum. Most time series have shorter gaps due to various reasons. Monthly and yearly averages from the
AMGI, HHI and IOF stations have been sent to the PSMSL since the stations were established.

Since 1950s, additional portable float-type chart-recording tide gauges were occasionally operational along the
Croatian coast of the Adriatic, with corresponding sea level measurements spanning over periods from 1 to 18 years, depending
on location: Vis and Ušće Neretve (measurements during 1957), Mali Ston (1957-1959), Broce (1957-1959), Ubli (1987-
1991), Sućuraj (1987-2005), Žirje (1989-1991), Zlarin (1983-1988), Gaženica (1983-1988) and Rijeka (1998-1999).

*Radar-type stations:* The Croatian tide gauge network was expanded in 2017-2021 with nine new coastal stations
based on radar technology. Stations in Stari Grad (Hvar Island), Sobra (Mljet Island) and Vela Luka (Korčula Island) were
installed in 2017 (within the framework of the UKF MESSI and MRRFEU POZOR projects) and are operated by the IOF.
Station Šibenik, located in Sv. Ante Channel was established in 2020 and is operated by the RBI. In 2021 the IOF established
stations Vis (Vis Island), Mali Lošinj (Lošinj Island), Bistrina (Mali Ston Bay), Raslina (Lake Prokljan - ocean station) as part
of the Interreg Italy – Croatia projects ECOSS, CHANGE WE CARE, and RESPONSe. Also, in 2021 the HHI deployed a
radar sensor in Rijeka near the traffic port. All instruments are installed in open air providing 1 min data which are averages
of 1 s measurements done for 20 s during each minute. All stations locally store these 1 min data and transmit them with up to
10 min latency to home institutions. The exception is Šibenik at which data is stored with a temporal resolution of 55 min.
Ancillary atmospheric pressure and wind are measured at Stari Grad, Vela Luka, Mali Lošinj, Vis and Bistrina (1 min data),
while surface current and sea temperature (depths 0.5, 1.0, 2.0 and 4.0 m) are measured at Šibenik station (every 55 min). Tide
gauges in Rijeka, Stari Grad, Vela Luka and Sobra are leveled to the national geodetic datum. By the end of 2022, stations
Rijeka and Sobra are planned to be upgraded with ancillary meteorological sensors. Vela Luka tide gauge was temporarily
decommissioned in June 2021, but it should be reinstalled during 2022. All non-quality-controlled sea level and atmospheric
data      from      the      IOF      stations      are      available      directly      through      the      IOF      website
(http://faust.izor.hr/autodatapub/mjesustdohvatpod?jezik=eng). In addition, sea level data measured at Sobra, Stari Grad and
Vela Luka are also available through the IOC-SLSMF website (since October 2018). Data from Rijeka and Šibenik are
visualized on https://adriaticsea.hhi.hr and https://hv.geolux-radars.com/sites/sibenik-svante.html, respectively.

In addition, the HHI plans to install two additional radar-type tide gauges in the following years in the areas of Šibenik and
Mali Lošinj. Stations operated by Croatian institutions are listed in Table A12.



### 2.1.13 Montenegrin tide gauge Network

The Institute of Hydrometeorology and Seismology, Department of Hydrography, is responsible for monitoring and maintenance of the tide gauges installed in the Montenegrin part of the Adriatic Sea. The network includes two permanent tide gauge stations based on float-type technology. The tide station in Bar was established in 1965 and till 1991 was the part of

Former Yugoslavia tide gauge network (Slovenia, Croatia, Montenegro). During the 1990s the tide station in Bar was not fully operational for a long time, but it has been restored, and re-connected to the national geodetic network through levelling. The second Montenegrin tide gauge station was established in Kotor in 2010. For both stations the sea level is measured once every 6 min, with GSM-based data retrieval to the central server. Stations operated by Montenegro are listed in Table A13.

### 2.1.14 Tide monitoring network in Albania

The Institute of Geoscience, Department of Hydrology, is responsible for monitoring all water resources in Albania, including the sea level. The sea level observations are taken manually, two times a day - at 7 am and 7 pm. These data are stored in a book by the observer and sent every month to the Institute of Geoscience (Tirana, Albania). At the centre, this information is archived and not controlled, unless there is a request for this data. Stations operated by Albania are listed in Table A14.

### 2.1.15 Sea level observations in the Maltese Islands

The routinely collection of sea level data in the Maltese Islands was initiated in May 1993 by the Physical Oceanography Unit (later Physical Oceanography Research Group) using an ENDECO-type 1029/1150 differential pressure gauge in Mellieha Bay on the northern coast of Malta. The station remained in operation until 2001, measuring in delayed mode every 2 min, supplemented with meteorological measurements collected at a nearby station in Ramla tal-Bir overlooking the southern Comino Channel. This endeavour was mainly intended to assess the sea level variability and to study the phenomenology of

strong seiches locally known as the 'milghuba' (Drago, 2000, 2009).

A MedGLOSS station installed in February 2001 in the Portomaso marina at the Malta Hilton in St. Julians constituted the first real-time ocean observing system in Malta. The instrument, donated by the International Commission for the Scientific Exploration of the Mediterranean Sea (CIESM), collected sea level data every 30 s, and also seawater temperature, atmospheric pressure and waves in the marina. Hourly averaged observations were shared in real-time with the MedGLOSS network

through the Israel Oceanographic and Limnological Research (IOLR) that coordinated the project. The system comprised an underwater Paroscientific pressure sensor, a type Digiquartz Intelligent sensor, and a Setra atmospheric pressure sensor. In 2010, the system was upgraded with new equipment to enable higher sampling rates, and to enable a first phase towards contributing to the Mediterranean Tsunami Warning System within NEAMTWS (The Tsunami Early Warning and Mitigation System in the North-Eastern Atlantic, the Mediterranean and Connected Seas).

More recently, in collaboration with the Joint Research Commission (JRC), the sea level observing network has been enhanced by two IDSL stations, set up at Delimara (March 2021) and Senglea (June 2021). Data is transmitted in real-time to



the University of Malta as well as to the JRC TAD server (https://webcritech.jrc.ec.europa.eu/TAD_server/Device/555) with a temporal frequency of 5 s. Each station is equipped with a radar sensor connected to electronics that also measures air temperature and captures visual images of the sea state every 15 min.

In March 2021, a Radac WaveGuide sensor was installed at the tip of the Marsaxlokk breakwater to measure sea level, wave height and wave period. The instrument was procured through the SIMIT-THARSY project, partially funded by the ERDF through the Operational Programme. This radar sea level gauge is capable of measuring water displacement with a resolution of 3 mm at a frequency of 10 Hz. The wave height is measured with an accuracy of 1 cm at 1 min intervals. In front of it, on the other end of the bay, another IDSL was deployed. The operational data from these stations is linked to other

observations and delivered in real-time through the PORTO stations web interface developed through the CALYPSO South project, another INTERREG V-A Italia-Malta project (www.calypsosouth.eu). Tide gauges operated by the Physical Oceanography Research Group of the University of Malta are listed in Table A15.

### 2.1.16 Greek tide gauge network operated by the Hellenic Navy Hydrographic Service

The Hellenic Navy Hydrographic Service (HNHS) monitors sea level variability over the Aegean and the Ionian Seas through

a network of 22 tide gauges. All hydrographic stations consist of analogue float type gauges equipped with rotating drums, rotating with speed of 1 cm/hr. These tidal stations are located within port facilities to record continuously sea level and provide these measurements to the relevant port authorities. The tidal stations at Thessaloniki, Kavala and Piraeus are the oldest of the network, operating continuously since 1933.

Since 1990, HNHS upgraded the network by installing digital water level sensors at seven selected stations (Piraeus,

Alexandroupolis, Kalamata, Katakolo, Lefkada, Siros and Chios), operating in parallel to the analogue systems. These stations collect, store and transfer water level data to central servers in near-real-time mode. Digital stations collect additionally a series of ancillary parameters, like the sea surface temperature and meteorological data (air temperature, humidity, barometric pressure and wind speed and direction). In parallel, during that period, the network's tidal data, recorded in paper charts, have been digitized into hourly values, and subsequently organized and stored in the HNHS databases. The water level error is

estimated to approximately 1 cm (Tsimplis, 1994).

Data from four stations (Piraeus, Katakolo, Siros and Kalamata) are visualised on-line through the HNHS web page (https://www.hnhs.gr/en/online-2/tide-graphs). Data from all tidal gauges covering the 1969–2020 period are also transferred to the Permanent Service for Mean Sea level (PSMSL) (https://www.psmsl.org/), supported by the International Oceanographic Commission (Bitharis et al., 2017). Sea level data from these four digital stations are also provided and

visualized in real-time mode to the IOC -SLSMF (http://www.ioc-sealevelmonitoring.org/). Stations operated by HNHS are listed in Table A16.



### 2.1.17 Bulgarian coastal sea level service

Systematic sea level measurements were initiated in Bulgaria in the beginning of the 20th century, with 16 tide gauge stations operational over a certain period of time till 2013. Operators of these sea level stations were: National Institute of Meteorology and Hydrology, Bulgarian Academy of Sciences (NIMH) – 6 stations, Cadastre Agency, Ministry of Regional Development and Public Works (CA) – 4 stations, Port Infrastructure (PI) – 5 stations and Institute of Oceanology, Bulgarian Academy of Sciences (IO-BAS) – 1 station. At present, five of these stations are operated jointly by the Institute of Oceanology, National Institute of Geophysics, Geodesy and Geography and the Geodesy, Cartography and Cadastre Agency and are providing data in real-time. The stations are equipped with a high accuracy radar instrument. Active Bulgarian stations are listed in Table A18.

### 2.1.18 National Institute for Marine Research and Development "Grigore Antipa" coastal sea level monitoring, Romania

Sea level measurements started in Romania in 1856, at the initiative of the European Danube Commission. However, these data have not been preserved, Regular recordings of the sea level started in Romania in 1933, by installing a float gauge. Presently, two methods of measurements are used in Constanta: a pressure sensor and a float gauge, with a hydrometric sight, at which visual measurements are performed three times a day for data quality purpose. The accuracy of the analogue measurements on a paper chart are estimated to 1 mm. At Sulina and Mangalia, the measurements are made using pressure sensors. The sea level data is transmitted to the server via GPRS/GSM method.

In Sulina, Mangalia and Constanta three IDSL were deployed in the last decade. Presently, only the one in Mangalia is operational. Stations operated by NIMRD "Grigore Antipa" are listed in Table A19.

### 2.1.19 Russian tide gauge network in the Black Sea

Sea level observations on the Russian coast of the Black Sea began in the middle of the 19th century. Systematic measurements of the sea level were initiated in 1873. Since 1944, the sea level network was restored, reconstructed and expanded. Sea level float-type recorders were installed at many tide gauges. Since 1977, all tide gauges have been tuned in a single system of heights (the Baltic Height System). In total, during the years, aat the territory of the USSR on the Black Sea, there were 44 tide gauges, the data from which were saved until 1985. For 23 tide gauges, there are long-term digital series of hourly observations of sea level. Today short-period sea level variations on the Russian coast are measured at five tide gauges: Tuapse, Sochi, Sevastopol, Yalta, and Feodosia. The Russian tide gauge network is owned by The Russian Federal Service for Hydrometeorology and Environmental Monitoring (Roshydromet) and is operated by the All-Russian Research Institute of Hydrometeorological Information - World Data Center and data availability. Tuapse tide gauge is co-located with a GNSS. Russian tide gauge stations in the Black Sea are listed in Table A20.





### 2.1.20 Coastal sea level monitoring in Türkiye

Sea level monitoring activities in Türkiye date back to the mid-1930s, when the first float-type tide gauge was installed at Antalya harbour to determine the national vertical datum. Since then, a considerable number of temporary mechanical and analogue tide gauges had been deployed at different spots. In 1999 the Turkish National Sea Level Monitoring System (TUDES) programme was initiated by the General Directorate of Mapping. Up to 2011, under this programme, a network consisting of 20 digital tide gauges with acoustic sounding tubes, for continuous measurements, was established. Due to the significant maintenance problems related to the acoustic gauges and the strong need for the VLM monitoring, all TUDES stations have been replaced with radar gauges after 2015, and most are GNSS collocated. The data averaged at 30 s and 15 min intervals are transmitted to the data center in Ankara in near-real time through GSM and internet. Real-time and delayed mode quality controls, data analysis, database management, and data distribution activities are performed at the data center. Further, TUDES is delivering sea level data to the regional networks (e.g. ICG/NEAMTWS). More information about the TUDES can be found at https://tudes.harita.gov.tr/?lang=us.

In addition, three IDSL stations were deployed in Bozcaada, Samsun and Bodrum. TUDES tide gauge stations are listed in Table A21, while in Table A22 are listed JRC IDSL stations in Türkiye operated by KOERI.

### 2.1.21 Cyprus tides gauges networks

The systematic monitoring and transmitting real-time sea level data in Cyprus was first initiated in 2001 as part of the MedGLOSS-Mediterranean sea level network. Initially, one tide gauge was deployed at Paphos harbour in 2001 by the Department of Fisheries and Marine Research Oceanography Unit (DFMR) and hourly data were transmitted along with atmospheric pressure and *in situ* sea temperature via internet to a dedicated MedGLOSS web page hosted by the Marine Data Center of the IOLR. Later, this station became part of the MedGLOSS/ESEAS (European Sea Level Service) and continued to transmit data until 2015. After the December 2004 catastrophic Indian Ocean tsunami, the Paphos sea level station was also set to transmit in real-time sea level data to the IOC-SLSMF every 1 min. Moreover, the station was included in the coastal sea level monitoring network of the NEAMTWS-North East Atlantic Mediterranean Tsunami Warning System (UNESCO, 2007).

In the frame of MedGLOSS/ESEAS activities, another three tide gauges were deployed in Cyprus during 2010, at the Zygi and Paralimni fisheries shelters and in the Larnaca marina. The tide gauge deployed at Zygi fisheries shelter was provided by the Cyprus Governmental Department of Lands and Surveys. These tide gauges were operated only for two years, between 2010-2012, the data of which, together with those of the Paphos station (operated between 2001-2015), were used to estimate the Lowest Astronomical Tide and the Lowest Low Tide at the locations of their deployment (Papazachariou, 2014). Furthermore, studies were carried out to examine the trend of the sea level variations at the Paphos station (Loizides et al, 2010), as well as inter-comparison of the Paphos time series data with satellite altimetry time series (Banks et al. 2003; Papazachariou et al., 2014)





The Paphos MedGLOSS/ESEAS tide gauge was equipped with a Paroscientific Digiquartz Intelligent pressure sensor pressure type, while the rest three Cyprus MedGLOSS/ESEAS tide gauges were equipped with AANDERAA pressure sensors. In the frame of the Interreg THALCHOR2 project, a new tides gauge network named PYTHEAS has been deployed in Cyprus during 2018 (Chris et al., 2020). The PYTHEAS tides gauges network consists of four stations (Paralimni and Pomos fisheries shelters and Paphos and Larnaca harbours), all owned by the Cyprus Governmental Department of Lands and Surveys (DLS), while the fifth one (old Limassol harbour) is owned by the Cyprus University of Technology. All the PYTHEAS sea level networks were equipped with radar tide gauge sensors, along with atmospheric sensors, such as pressure, air temperature, humidity, as well as sea temperature. The station at the old Limassol harbour was also set to serve as a GNSS reference. The data from the PYTHEAS tides gauges network are transmitted in real-time to the hydrographic database of the DLS.

In parallel and independent from the PYTHEAS network, an additional acoustic tide gauge station was deployed at the end of March 2018 by the European JRC-Joint Research Center at the Zygi fisheries shelter, next to the location of the Cyprus MedGLOSS/ESEAS Zygi station. The 5 s data from this station are provided in real time also to the IOC-SLSMF (http://www.ioc-sealevelmonitoring.org/station.php?code=zygi1).

In 2019, an offshore station for sea level variation was deployed in the frame of Interreg HERMES project using a bottom mounted AWAC ADCP (Acoustic Current Doppler Profiler) pressure sensor. The ADCP was deployed at the water depth of 40 m in the Larnaca Bay, close to the famous scuba diver shipwreck "Zenobia". The ADCP measures in addition to the sea level variation, sea currents at 20 water depths, waves, sea temperature and suspended particles. The data are transmitted in real time via cable from the bottom mounted ADCP to a connected surface oceanographic buoy, and then via a GPRS to the ORION NGO server for use in the dedicated HERMES web page (Zhuk et al., 2020). Cyprus stations are listed in Table A23.

### 2.1.22 Egyptian tide gauge network

Six tide gauges were operational along the Egyptian Mediterranean coast. These gauges were deployed at Port Said, Burullus new harbour, Abu-Qir Bay, Alexandria Western Harbour, Sidi Abdel-Rahman and Mersa Matrouh. The periods of data availability are different for each location, with the longest records (30 years) at Alexandria Western Harbour and the shortest records (4 years) at Mersa Matrouh.

In June 2018, in collaboration with the JRC, the National Institute of Oceanographic and Fisheries deployed an IDSL station in the port of Alexandria. Stations of Egyptian tide gauge network are listed in Table A25.

### 2.1.23 Algerian tide gauges network

The National Institute of Cartography and Remote Sensing (INCT) is responsible for equipping the national territory with all kinds of geodetic networks: GNSS, gravity and levelling networks. Historically, the altitudes of the Algerian levelling network are reported to several origins generally chosen in an arbitrary manner, level deduced from the indications of the Medimaremeter of La Goulette (Tunisia), altitude of the landmark of Porte De France (Tunisia), or the coast of La Goulette (Tunisia), console placed at Sidi El Hemessi station (Tunisia) in 1914.





Being aware of the significance in providing the national territory with a precise altimetric reference, the INCT put the effort in the installation of automatic tide gauges along the Algerian coasts, first to bring the bathymetric surveys to a stable reference, hydrographic zero or nautical chart zero, and then to predict the tide or define reference levels. In addition, the INCT is in charge for setting up the hydrographic zero as an altimetric reference for water heights, and uses any tidal data as a national referent (official journal): general knowledge of the tide, determination of harmonic constants and extreme levels and tidal prediction.

The installation of six tide gauge stations with automatic acquisitions along the Algerian coasts (Ghazaouet, Oran, Ténès, Algiers, Jijel and Annaba) in the mid-2010s upgraded substantially the network, in particular for monitoring sea level variations and modernization of the national altimetric reference. Currently, the INCT is additionally upgrading the network through setting up permanent GNSS stations in collocation with the tide gauges stations at the ports of Algiers, Jijel, Oran, Annaba, Ghazaouet and Ténès. This approach would constitute an important phase for the creation of a multi-observation observatory for spatial measurements, gravity field, levelling and tide gauge. Upgrade of observatories to have real-time data acquisition is also planned for near future. Stations of Algerian tide gauge network are listed in Table A26.

**2.2 Summary of the existing coastal sea level monitoring infrastructure**

With the aim of facilitating a more general assessment of the coastal sea level monitoring networks individually described in Section 2.1, a brief questionnaire survey was conducted, resulting with information on: i) institution, country, contact name of a network; ii) number of stations; iii) main purpose of the network; iv) funding mechanism; v) data policy; vi) raw time sampling interval; vii) data latency; viii) number of sea level sensors at each station; ix) type of tide gauge; x) levelling strategy; xi) number of tide gauges collocated with a GNSS; xiii) ancillary measurements; xiv) quality control and processing and xv) public data availability. The survey responses are tabulated and may be found in Supplementary material.

Based on the responses received, 236 active stations have been identified so far in the M/BS. These are operated by 30 agencies, representing all countries in the region except Morocco, Tunisia (both have sea level networks, e.g. Jabnoun and Harzallah, 2020) and Libya. We could also not get information from the Italian Hydrographic Office, that operates Genoa tide gauge, one of the longest-operating Mediterranean tide gauges. In some countries tide gauges are operated by several institutions (e.g. 6 in Spain, 5 in Croatia), while others have one national network managed by a single agency (e.g. SHOM, in France). ISPRA in Italy runs the largest network in terms of number of stations (62), followed by the JRC (24 tsunami stations operated jointly with other institutions), HNHS in Greece (23) and the TUDES Turkish network (20). The majority of respondents are responsible for a smaller number of stations, even for just one single station in several cases.



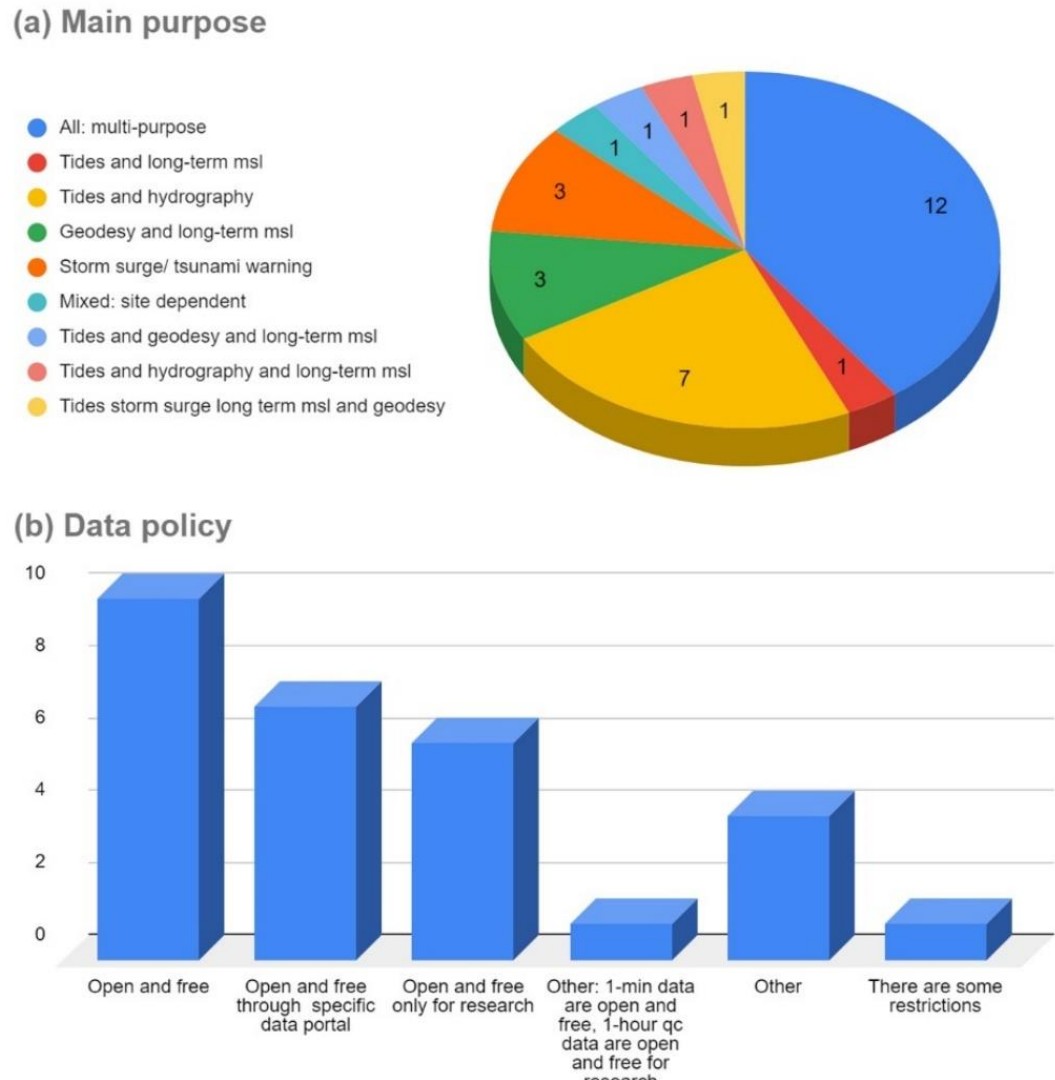

**Figure 2. (a) Primary objective of the coastal sea level monitoring networks based on responses to the survey, and (b) data policy of the operating institutions as collected by the survey.**

Figure 2a displays the primary objective of the networks, chosen by operating institutions between the following non-exclusive choices: i) tides and hydrography; ii) storm surge/tsunami warning; iii) geodesy and long-term MSL; iv) tides and long-term mean sea level (MSL); v) altimetry calibration; vi) models validation; vii) harbour operation and aid to navigation; viii) meteotsunamis or xi) multi-purpose (all previous options). Most of the networks, especially the largest ones, are claimed to be multi-purpose (40% of answers:12), followed by those mainly aimed at tides and hydrography applications (23% of the





answers: 7), such as the HNHS Greek network, Romania and one of Cyprus networks. Three institutions reported that their stations are focused on storm surge/tsunami warning and another three on geodesy and long-term MSL. None of the stations

have been specifically implemented for applications v)-viii). The IOLR network in Israel provided a mixed, site-dependent answer, depending on technology and time sampling and latency (e.g. the 2 radar sensors installed for storm surge/tsunami warning). Despite more than half of the answers claim open and free access to data (altogether 10 agencies, and 7 agencies through specific data portals), there are also some data policy issues (Fig. 2b): open and free only for research were selected by 6 respondents (+1 for 1-h qc data), while 5 institutions mentioned unspecified issues or restrictions on data access.

With respect to network sustainability related to maintenance strategy and status, most of the respondents confirm no problems of funding now or in the near-future. Problems with maintenance are reported by NIOF in Egypt, and for a couple of stations in Croatia (IOF) and Israel (IOLR). Most serious issues are raised by the Albanian network, which is not being maintained at this moment and there are no plans for funding in a short term. JRC also warns that the network funding is guaranteed now, but it might stop anytime. All the agencies rely on their own resources for *in situ* maintenance except PdE

(Spain), ISPRA (Italy) and Croatian Waters (Croatia), that subcontract this work.

Raw time sampling interval is 1 min for most of the networks in the region (12 of the respondents). However, a large range of raw sampling options are provided by the rest of contributors, from 6.7 Hz resampled to 5 s (JRC tsunami stations) to 55 min (Ruder Boskovic Institute), 1 h (Croatian Waters) or 2 h (Meteolestartit). Some agencies are more specific in the answer to this question and report several samplings available depending on data portal or application. Latency of data

transmission is claimed to be real time (<=1 min) by 9 of the respondents (IOLR only for the two radar sensors), near-real time (minutes, hours or days) by 15 of them and delayed mode access only by 7 agencies.

One of the most challenging aspects of a tide gauge network is datum stability and link to other official references, especially for long-term and mean sea level studies. According to the responses, the majority of networks perform high-precision connection of the tide gauge benchmark (TGBM) to the national geodetic datum, as the basic levelling strategy. In

some cases, they provide the date of last levelling campaigns, but in general there is a lack of information about their frequency, apart from periodic levelling near the TGBM during routine maintenance reported by 4 institutions (usually once per year). Connection of all stations to a nearby GNSS station is reported by the Spanish IGN and by the Algerian network.

With respect to quality-control and data processing, the response is also diverse, as shown in Fig. 3. In most cases only delayed-mode quality-control is performed by data originators (17 respondents, 6 of which also generate sea level

products). In addition to delayed-mode quality-control and products, automatic quality-control in near-real time is also performed nowadays by 7 agencies in the area, while only 2 agencies do not perform quality-control or processing at all.





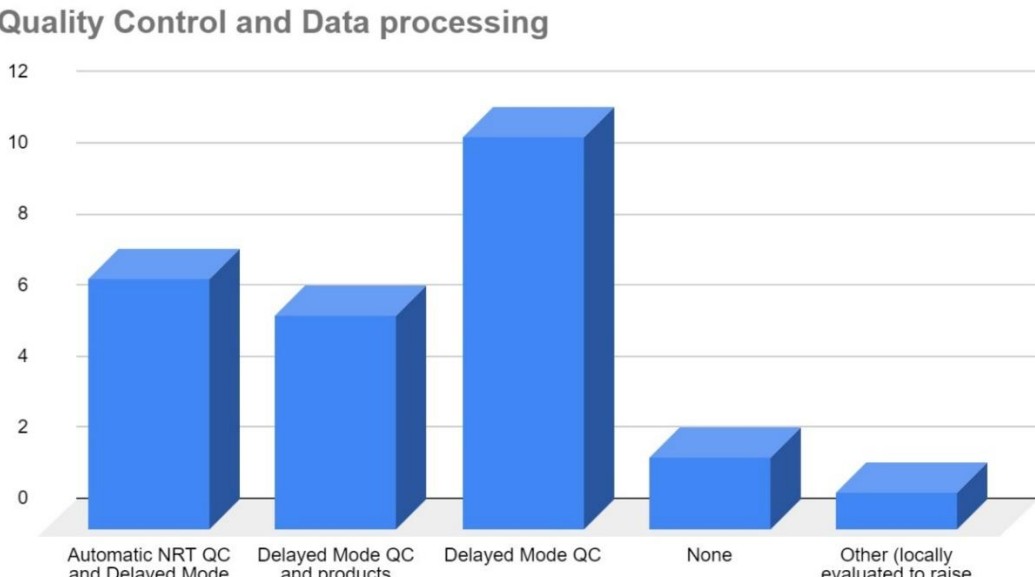

**Figure 3. Quality-control and data processing strategies followed by M/BS tide gauge network operating agencies.**


The list of stations including name, coordinates, data period and ancillary measurements is provided in Appendix (Tables A1-A27). This list is the basis for detail mapping of existing sea level infrastructure in the M/BS, composed of 236 active stations, including type of sensor (radar, acoustic, float or pressure sensor), ancillary meteorological or oceanographic

data, or collocation with a nearby permanent GNSS station (Figs. 4-6). The data periods on this table reveal that the oldest active stations in the M/BS are: Marseille: 1849 - present (France), Trieste Molo Sartorio: 1875 - 1889; 1901 - present (Italy), Burgas port: 1910 - present (Bulgaria), Varna port: 1919 - present (Bulgaria), Venice Punta della Salute: 1924 - present (Italy), Alicante 1: 1928 - present (Spain) and Bakar: 1929 - present (Croatia). These stations, the most relevant for climate research, started with float gauges and have been upgraded to radar sensors in most cases. Very often, time series from different tide

gauges at the same harbour are used to generate a dataset spanning the whole history of the station (e.g. new Alicante dataset by Marcos et al., 2021, starting with first measurements by a tide pole in 1870).

In terms of technology, Figs. 4-6 reveal that today the radar sensors are the most common type of instrument in the network, being sea level sensor at 134 stations (57%), dominating especially in the Western Mediterranean. Further, 62 stations (22%) are still based on, or combined with, traditional float gauges, mainly along the eastern Adriatic coast and in Greece.

Acoustic sensors are used at 27 locations (mostly tsunami sensors), while pressure sensors are used at 15 stations (Balearic Islands, Western Black Sea and Israel). At four stations sea level measurements are still performed manually using a tide pole (Albania).




Only 6 stations that have two different sensors: float and radar (Koper and Bakar), float and pressure (Constanta), float and acoustic (Algiers, Jijel and Oran). GLOSS recommendation for tsunamigenic areas is to use a radar plus a pressure

sensor, a combination not found in the M/BS. A promising news is the increasing number of stations collocated with a GNSS, and therefore with potential geocentric reference (TGBM ellipsoidal height) and VLM information: 46 locations (19% of the network): 26 in the Western Mediterranean, 2 in the Central Mediterranean, 9 in the Eastern Mediterranean, 3 in the Marmara Sea and 6 in the Black Sea. At 150 stations (64% of active stations) there are ancillary sensors which provide meteorological data (90 stations, mainly atmospheric pressure and wind), oceanographic data (11 stations) or both (49 stations, many of them

in the Central Mediterranean).

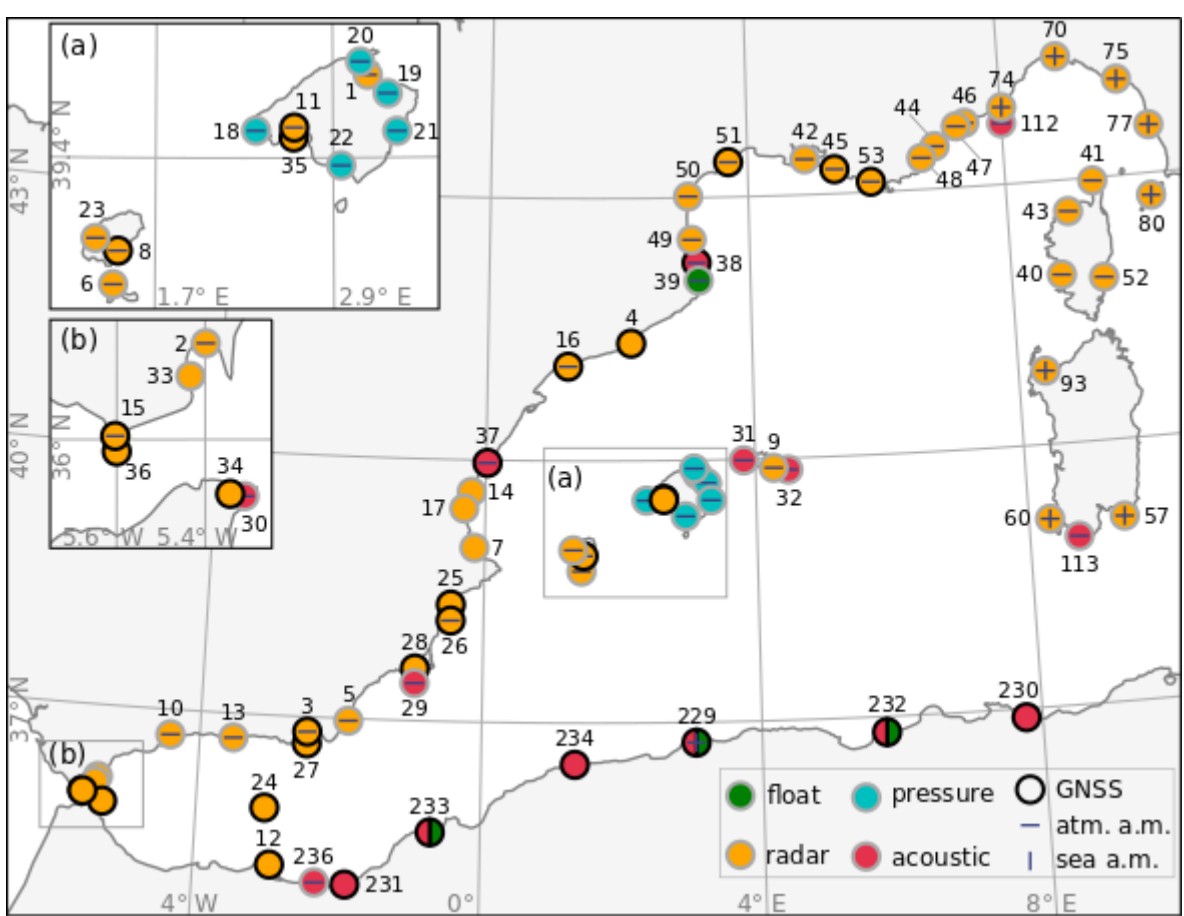

**Figure 4: Active tide gauge stations in the Western Mediterranean Sea. Marker colour indicates tide gauge sensor, while black circle indicates collocation with GNSS station. Ancillary measurements, atmospheric or oceanographic, are indicated with navy-blue**
**horizontal and vertical dashes, respectively. Station names are listed in the Appendix tables: Tables A1 -- A9 and Tables A26 -- A27. Stations 26, 27, 29, 32, 34, 35, 36 and 112 are slightly shifted to assure better visibility.**





**Figure 5: Same as in Figure 4, but for the Central Mediterranean Sea. Station names are listed in the Appendix tables: Table A8 and Tables A10 -- A17. Stations 82, 103, 114, 121, 123, 124, 126, and 128 are slightly shifted to assure better visibility. Some stations appear to be inland due to the relatively coarse resolution of the coastline.**





**Figure 6: Same as in Figure 4, but for the Eastern Mediterranean Sea and for the Black Sea. Station names are listed in the Appendix**
**tables: Tables A13 -- A14, Tables A16 -- A25, and Table A27. Stations 137, 152, 156, 165, 168, 170, 172, 178, 201 and 218 are slightly**
**shifted to assure better visibility.**

### 3 Data availability in existing international programs

In this chapter, we assess the M/BS sea level data availability. The assessment is primarily based on the public global and

European data repositories as of 15 November 2021, which have been established for various applications (from real-time

applications to research-quality products) and contain data of different time sampling step (from a minute to a month). Figures

7 and 8 summarize information on the presented data repositories.



## 3.1. Monthly sea level data repository

The oldest sea level data repository, at which monthly sea level data have been collected for almost a century, is the Permanent Service for Mean Sea Level (PSMSL, www.psmsl.org). The PSMSL was established in 1933 by Joseph Proudman, with an
aim of collecting, analysing and interpreting monthly sea level data at the global scale (Woodworth and Player, 1993; Holgate et al., 2013). The collected data is of high quality and freely available, thus providing an excellent dataset for mean sea level studies in this era of climate changes (e.g. Spada and Galassi, 2012). In total, there are 158 M/BS data series with a high reliability (classified as revised local reference (RLR) data) (Fig. 8a), with six records spanning over more than 100 years. These stations are Trieste, Marseille and Genova in the Mediterranean, with the respective length of 146, 136 and 114 years
and data coverage of 87%, 97% and 78%, then Poti, Batumi and Tuapse stations in the Black Sea (145, 137 and 103 years of length, respectively, and data coverage of 94%, 87% and 99%, respectively). Due to their exceptional length, these series have been thoroughly checked (e.g. Tsimplis and Spencer, 1997; Wöppelmann et al., 2014) and used in numerous long-term sea level assessment studies (e.g. Letetrel et al., 2010; Pashova, 2012).

In the PSMSL repository, there are 38 tide gauge stations with monthly records longer than 50 years, concentrated
mostly along the northern coastlines of the M/BS (Fig. 8a). Along the North African coast there are long-term (50+) data from only two stations, Alexandria and Ceuta, making proper quantification of sea level changes along the southern Mediterranean challenging (e.g. Gomis et al., 2012). At a number of sites (24 of them) there are two or more tide gauges collocated, with the long-term data coming from the float-type tide gauges, and with additional sea level series normally coming from the digital instruments (radar, acoustic or pressure tide gauges) and spanning over the last few decades at best. The median data coverage
of the M/BS PSMSL series is 92%, while 63 stations have the data coverage higher than 95%. There are 19 sea level records which have 100% data coverage, but all of them are relatively short records (up to 30 years) – these series mostly come from digital instruments. Oppositely, there are 4 sea level records with data coverage less than 50%, with 3 of them containing data from the 1960s and 1970s.

## 3.2. Hourly sea level data repositories

At hourly resolutions, some of sea level records coming from the PSMSL tide gauges are also available in the Copernicus Marine Service (CMEMS) In Situ Thematic Assembly Centre (http://www.marineinsitu.eu). In general, the CMEMS is a service providing operational data-driven ocean products, plus systematic information on the ocean and sea-ice state (Le Traon et al., 2019). The CMEMS database contains hourly sea level records from 115 tide gauges in the M/BS (Fig. 8b), with the median length of the series of 2 years. Hourly sea level data records can also be found in the EMODnet Physics *in situ*
observations repository (Novellino et al., 2015, https://www.emodnet-physics.eu/map), which is developed to be a single point of access to near real time and historical ocean data in Europe. The repository contains hourly sea level observations from 119 tide gauges in the these two basins (Fig. 8b). For some stations, the data is available both from recent observations and from historical periods, sometimes of the same length or even longer than the PSMSL records, but mostly covering shorter periods





and sometimes without any data available for download – these "empty" stations are not included here. There are 12 hourly

sea level series in the EMODnet repository longer than 30 years, while more than half of stations have records of 4 years long

or shorter.

In conjunction with EMODnet Physics and CMEMS, the relevant research product for the M/BS, containing sea level

data at hourly resolution, is Global Extreme Sea Level Analysis dataset (GESLA, http://www.gesla.org, Woodworth et al.,

2016, 2017) - here version 3 is analysed. GESLA is a research-driven sea level product containing series of hourly or higher

sampling resolution, which encompass 1355 sea level records and 39151 station-years all over the world. In the M/BS, GESLA

lists 101 stations, most of them spanning over periods much shorter than listed in PSMSL, up to a few decades. The longest

GESLA record in the Mediterranean is the one for Marseille (France) tide gauge, 173 years long, while the median length of

GESLA records is 15 years.

In addition to the quoted centralized data repositories, some of hourly sea level data can also be accessed through

SeaDataNet portal (https://www.seadatanet.org), which is a virtual centre that provides different ocean metadata, data and

products archived by data providers. However, the access to the sea level archives is not straight-forward. It is mostly

distributed towards data providers, sea level data is often combined with data from other observing platforms (e.g.

echosounders, synthetic aperture radars) – data is also provided in different formats and split in small observing intervals –

making this portal rather complicated to use. Still, some data not listed in the EMODnet Physics, CMEMS and GESLA dataset

may be found there, like sea level records for the Georgian, Ukrainian and Maltese stations.

It is a challenge to a researcher to choose the best database out of the three and to locate research-quality data. In

various databases, data originating from the same location frequently have different names, identification numbers and

metadata. We thus encourage founders of these data repositories to join their data together, and provide one high-quality dataset

for further research, with unique data policies allowing for accessibility of the data and following the FAIR (Findable,

Accessible, Interoperable and Reusable) principles (Wilkinson et al., 2016).

### 3.3. Minute sea level data repositories

Another important sea level data portal is the UNESCO Intergovernmental Oceanographic Commission Sea Level Station

Monitoring Facility (IOC SLSMF, http://www.ioc-sealevelmonitoring.org), which – contrary to other sea level data

repositories – has not been developed for collection of the research quality data, but for operational purposes (Aarup et al.,

2019). The IOC SLSMF has been developed to deliver the information about the status of tide gauges operating in real time,

as well as to visualise the data downstreaming to the service (Flanders Marine Institute (VLIZ), 2021). Further, the service

provides high-resolution (mostly with a minute resolution) sea level data at global level, following demands of the operational

and early warning systems that emerged after the 2004 Indian Ocean tsunami.

As of 15th of November 2021, the IOC SLSMF provides information and data from 143 tide gauge stations in the

Mediterranean and Black Sea, most of which (all but 30 for the preceding week) are operational in real-time with data

transmitting latency between 1 and 10 min (Fig. 8c). Ten stations have been operational since 2008, when they were installed





as a result of activities of the IOC Intergovernmental Coordination Group for the Tsunami Early Warning and Mitigation System in the North-Eastern Atlantic, the Mediterranean and Connected Seas (IOC ICG NEAMTWS, Amato, 2020), established to coordinate regionally the development of the tsunami early warning system. The most recent additions to the 765 network are from 2019 and 2020, and are related to a few Spanish, Greek and Turkish tide gauges. The availability of the IOC SLSMF data in the Mediterranean is largely dependent on the data providers, which normally upgrade large segments of networks at once. E.g., most of the Italian sea level records have been available from December 2013, which is also the median value for the beginning of data provision of all the Mediterranean IOC SLSMF stations (Zemunik et al., 2021b). Most of the stations from which data are available at the IOC SLSMF database, are equipped with the radar tide gauge technology, 770 sampling rates are normally 1 min, except for Greek and Turkish sea level records that are largely available with the resolutions of 30 s and 20 s.

Recently, European Commission through the Joint Research Centre (JRC) at Ispra established a sea level database (https://webcritech.jrc.ec.europa.eu/, here marked as JRC-SLD, Fig. 7), containing multi-year records on a minute up to a ten minute resolution. The database assembles 105 stations in the M/BS operating in real-time, providing also the estimates of 775 tidal constants for each site. In addition, the companion site at https://webcritech.jrc.ec.europa.eu/tad_server/ is collecting the sea level data at a rate of 5 s, aimed to improve the tsunami hazard monitoring, in particular in the Mediterranean Sea and in the North Atlantic area (NEAMTWS area of UNESCO IOC).

As no global research-quality sea level data repository containing the data at a minute resolution has been developed yet, Zemunik et al. (2021b) applied quality-check procedures on the IOC SLSMF data and provided the Minute Sea level 780 Analysis product (MISELA), to be used for researching of high-frequency sea level phenomena. MISELA dataset contains 331 sea level records and 2303 station-years between 2004 and 2019, with resolution of 1 min and containing only high-frequency part of the signal (cut-off period at 2 h). In the M/BS, MISELA quotes 36 stations, covering a time period from 2008 at the earliest up to 2019, with the longest record coming from Melilla and Barcelona, Spain.

### 3.4. Data-providers' sea level data repositories

Aside from the listed global sea level data repositories and research products, these sea level data is also accessible through websites of data providers (e.g. PdE, SOCIB, SHOM, ISPRA; see more in Section 2). Further, there are sea level data that are not included in listed repositories, like ISPRA Rete Mareografica Laguna di Venezia, containing sea level records from 24 tide gauge stations, but are available through local repositories (https://www.venezia.isprambiente.it). Last but not least, there are more sea level records, in particular at hourly timescale and the raw data, not easily reachable by users. Some 790 of these tide gauge operators provide the access only to recent sea level data, some are visualised through a graphical interface, while others are available on demand – often only for research purposes, and other have even more restrictive data policies.




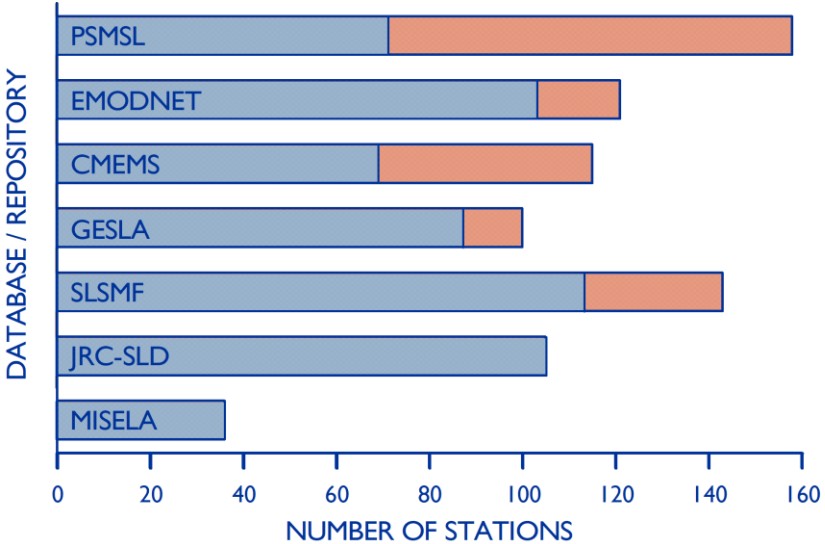


**Figure 7. Number of tide gauge stations in the Mediterranean and Black Sea for which the sea level data can be found in the quoted databases or data repositories. As of 15 November 2021, the stations having no data since 2018 for PSMSL, since 2021 for EMODNET and CMEMS, since 2020 for GESLA and being not operational in the preceding 7 days for SLSMF and JRC-SLD are indicated by red. For MISELA data is available up to 2019, at best.**








**Figure 8. The Mediterranean and Black Sea data accessible as: (a) monthly sea level averages in the PSMSL (the length of series is indicated by the size and colour of the circle; thick border lines denote the series that end in 2018 or later, and grey the series that end earlier, (b) hourly sea level data in either EMODNet Physics, CMEMS and GESLA (the length of series is indicated by the size and colour of the circle; thick and grey lines denote the series that do or do not, contain the data in 2021, respectively), and (c) minute series in the IOC SLSMF (the length of the operability period is indicated by the size and the colour of the circle; thick and gry border lines denote the stations that were or were not, operational on 15 November 2021, respectively).**





## 4 Assessment of the network to fulfil targeted applications

Based on the information compiled in Sections 2 and 3, an overview of the fit-for-purpose status of the network, to fulfil some specific applications, is presented below. Examples of use of tide gauges in the M/BS are provided, focusing on early warning systems, long-term sea level variability or altimetry calibration, all in order to show the main gaps or lack of information in terms of coastal in-situ measurements.

### 4.1 Operational, early warning and forecasting systems

**4.1.1 Tide tables and port operations**

Historically, the first requests for sea level information came from the needs of navigation near the coasts and access to ports with significant tidal ranges. Tide scales or other visual systems quickly gave way to tide gauges, which become a standard in port operations, in particular at these ports with intense traffic and large tidal range (Lemon, 2003). Although the traffic in the M/BS is substantial, the tides are small and ranging normally to a few tens of centimetres, except in the northern Adriatic and

Bay of Gabes (Tsimplis et al., 1995). Nonetheless, several countries produce tide tables to predict the astronomical tide.

The tides are normally predicted using harmonic analysis of measured signal. Sea level observations allow for estimation of the harmonic constituents necessary to recompose the tidal signal and provide highly precise estimates of the tidal signal. In the Tide Table, predictions of the times and heights of high and low tides are published for certain points, called reference ports or main ports, for which there are generally long series of observations that allow to obtain an accurate

prediction. Each reference port can be linked to one or more secondary ports. For these secondary ports, predictions are obtained by applying time and distance offsets from the reference port.

According to the International Hydrographic Organization (IHO) recommendations, the reference level of the tide/water height observations and predictions for navigators should be the same as the chart datum.

Use of tide gauges for computing tidal harmonic constants and tide predictions, usually done by national hydrographic

services, has allowed port authorities along the world to programme their operations well in advance. This is particularly important for those ports that receive large cargo ships with a large draft, for which the time of high or low water is critical. In the M/BS, where the tide is often very small, even negligible, tide tables are only used on those areas with higher tidal ranges. This is the case of the northern Adriatic Sea, where port authorities and pilots in Croatia, Slovenia and Italy use Tide Tables with this purpose. Tide predictions are not used at some other coastlines with smaller tides: as an example, the Spanish

Hydrographic Office does not generate Tide Tables for Spanish Mediterranean ports. In these cases, shorter-term sea level forecasts based on storm surge models are the main prediction tool available for harbour operators, as it is the case of Barcelona harbour, that relies on the forecasting system developed by PdE and described in Section 4.1.2. Of course, more precise information can simultaneously be derived from sea level observations transmitted in real time by the tide gauges.

As ships are getting larger and larger, they have more and more drafts, leaving little room for manoeuvre for

navigation. The precise knowledge of the water level in real time allows the port authorities to optimize the movement of ships.





PdE in Spain has developed a harbour visualization tool essential for managing port operations, which integrates real time tide gauge data with other measurements (e.g. wind) and data from models. Sea level is a key component of a local early warning system for each harbour, which issues alert messages whenever sea level or high-frequency sea level oscillations over predetermined thresholds occur. These thresholds can be configured by harbour operators. In Croatia, the Hydrographic

Institute also developed a web application with real time and predicted data, so port authorities and pilots can visually see the difference and adjust their operations accordingly.

Real time tide gauge data are also used by the ports during local bathymetric surveys and dredging activities.

The data collected by the sensors all around the world are used by the JRC Sea Level Database (JRC-SLD, https://webcritech.jrc.ec.europa.eu/) to compute the harmonic coefficient needed to forecast the tide in locations of the IDSL

sensors and the surrounding areas. Most of the data collected by the JRC-SLD are analysed with the same software used by the IDSL network to assess the behaviour of the sea level rise and decrease. In case of anomaly, this detection disseminates alerting information.

**4.1.2 Storm surges and coastal flooding**

Apart from high-frequency sea level disturbances (discussed separately in Section 4.1.3), the Mediterranean coastal floods

typically occur due to constructive superposition of tides and meteorologically induced storm surges. High local precipitation, increased river discharge and waves may further worsen coastal flood risks and lead to compound flooding episodes (Bevacqua et al., 2019). All these events pose a threat to cultural heritage, and densely populated coastal communities of the M/BS.

Storm surges are most severe along Tunisian, Aegean and Adriatic coasts, at which the highest number of sea level extreme events per year is observed (Cid et al., 2016). Especially critical is the area of the northern Adriatic Sea and Venice

Lagoon, exposed to the well-known a*cqua alta* phenomenon. To face these hazards, coastal regions rely on forecasting operational systems (Umgiesser et al., 2021), measurement networks and extreme events research activities (Calafat et al., 2014). In the Venice Lagoon, investment in coastal protection through planning and building of flood barriers has been essential.

These challenges are purely operational: short-term day-to-day flood risk needs to be continuously estimated on

synoptic timescales using observation-driven numerical forecasting models and other types of early warning systems (Žust et al., 2021; Makris et al., 2021; Ferrarin et al., 2020; Bajo et al., 2019; Mel and Lionello, 2014; Ferrarin et al., 2013; Pérez Gómez et al., 2012, 2021). Ensemble modelling allows dealing with uncertainties by generating probabilistic envelopes of possible sea levels (Bernier and Thompson, 2015; Bertotti et al., 2011, Ferrarin et al., 2020). Of course, different modelling, ensemble and observation-ingestion paradigms, may be used (Calafat et al. 2014). Panoramically speaking, modelling of sea

levels can be classified into numerical physical models on one hand and deep learning approaches on the other. Advantages of the former are typically more extensive spatial coverages - which come at a higher computational cost - and better performance in the extreme tails of sea level distributions. Deep learning systems on the other hand demonstrate rapid progress in their forecasting capabilities at very low computational cost (once trained) (e.g. Žust et al., 2021).



Operational forecasting sea level models, producing forecasts on the timescales of hours to days, typically cannot
resolve low-frequency sea level variability on the timescales of days to weeks (shown to significantly precondition coastal
flooding in some parts of the Mediterranean; e.g. Pasarić et al., 2000; Pasarić and Orlić, 2001; Ferrarin et al., 2021) and are
further constrained by the limited sea state knowledge at the time of the simulation. Therefore, sea level observations must
somehow be introduced into the model during or after runtime. This can be done in several ways. A simple nudging scheme
makes use of near-real time tide gauge data from the last 7 days in the Nivmar storm surge forecasting system run by Ports of
Spain since 1998 (Álvarez Fanjul et al., 2001). Another reliable approach to do this is tide gauge data assimilation (*e.g.* using
ensemble Kalman filtering as in Bajo et al., 2019), followed or complemented by statistical bias corrections with reference to
real-time sea level data. Deep-learning approaches (e.g. Žust et al., 2021; Bajo and Umgiesser, 2010) on the other hand offer
an alternative way of handling sea level observations: providing near-real-time sea level observations together with tidal model
forecasts enables deep networks to learn the biases in tidal models and compensate for this in real time without resorting to
numerically expensive schemes like data assimilation.

The Adriatic Sea serves as an example where observation-driven ensemble sea level modelling is absolutely
imperative for reliable predictions of cyclone-induced wind driven floods in Venice and other coastal towns along the northern
Adriatic coast. Its elongated shape leads to well defined seiche periods and resonant amplification of tides (Medvedev et al.,
2020). Its bathymetry on the other hand leads to topographic amplification of sea level signal on the northern Adriatic shelf.
Total sea level signal in the northern Adriatic therefore critically depends on the mutual reinforcement between storm surge,
tides and seiches, which in turn depends on the temporal phase difference between peak storm surge, peak tide and peak seiche
(see e.g. Cavaleri et al., 2010). High sensitivity to the phase difference between these components is the reason that even minor
errors in predicting the storm timing or trajectory may lead to substantial errors in the total sea level forecast (Cavaleri et al.,
2020), which makes ensemble modelling a clear advantage.

Focusing on short term forecasting of the Venice Lagoon sea level, ISPRA developed and manages an integrated
system, made up of *in situ* data coming from the RMN and RMLV networks (see Section 2.1.8) and numerical and statistical
models. The sea level forecasting system is mainly based on the deterministic hydrodynamic finite elements numerical model
SHYFEM, and it provides sea level forecasts up to 96 h depending on the spatial resolution (40 km in the Mediterranean Sea,
2 km in the Adriatic Sea, 100 m in the Venice Lagoon). It uses ECMWF and BOLAM meteorological fields, as input data,
and it assimilates the sea level measured by the 36 RMN tide gauges. This integration and the improvement of both *in situ*
observations and modelling system shows a virtuous example of efficiency and functionality to prevent and mitigate the impact
of flooding and meteo-marine extreme events on the Italian coastal environment.

Other operational storm surge forecasting systems have been in place in the Mediterranean Sea for the last two
decades (Umgiesser et al., 2021). These systems are progressively being improved or combined with existing 3D baroclinic
models, new higher resolution models, and ensemble and multi-model statistical techniques that provide sea level forecasts
with a confidence interval (probabilistic forecast). In the Western MS, Ports of Spain runs a multi-model storm surge forecast
(named ENSURF: Ensemble SURge Forecast: Pérez González et al., 2017, Pérez Gómez et al., 2021) that combines the output



of the abovementioned storm surge forecasting system Nivmar with CMEMS circulation models today operational in the region (IBI-MFC (Sotillo et al., 2015) and MED-MFC (Clementi et al., 2019)). The system employs the Bayesian Model Average (BMA) statistical technique, for which near-real-time data from the 17 REDMAR tide gauges in this coast (Section 2.1.1) are used to generate an improved probabilistic forecast at these harbours. This technique is also a valuable tool for operational validation and detailed assessment of the different operational systems.

A multi-model ensemble forecasting system has also been recently developed for the Adriatic Sea combining 10 models predicting sea level height (either storm surge or total water level) and 9 predicting waves characteristics (Ferrarin et al., 2020).

On longer timescales, mid- to long-term coastal management and spatial planning indicates a growing need to understand the impacts of climate change and global mean sea level rise on coastal floods in terms of their intensity and frequency on multi-decadal timescales (Međugorac et al., 2021; Oppenheimer et al., 2019; Bonaldo et al., 2019; Lionello et al., 2017; Androulidakis et al., 2015; Šepić et al., 2012; Marcos et al., 2011). Multi-decadal time series of sea level observations are indispensable in available research since they are the only way of pinning model reanalyses (*e.g.* Escudier et al., 2020) to past sea level variability, thus paving the way for reliable projections.

Several long-term studies (Androulidakis et al., 2015) indicate that different dynamic contributions to the global mean sea level will have to some extent compensat each other, but it can nevertheless be claimed with very high confidence that global mean sea level rise will lead to a substantial overall risk increase in coastal extreme events (Oppenheimer et al., 2019). Consequently, a well-functioning sea level observation network will be more and more imperative for synoptic coastal flood forecasting and mitigation in the M/BS.

### 4.1.3 Tsunamis and meteotsunamis

A carefully planned network of real-time accessible tide gauge stations is a must for efficient research, monitoring and issuing of tsunami and meteotsunami early warnings. The meteotsunami network should, in addition, be supplemented with air pressure and wind sensors.

Tsunamis, as earthquakes, cannot be predicted. Once they are triggered, mostly by submarine earthquakes, they propagate over thousands of km in the ocean and will reach the coastline in a matter of hours, or even of minutes for those coastal areas which are closer to the tsunami source. As fast detection is essential, tsunami warning systems must rely on real-time seismological networks (real-time information about the earthquake), and real-time information of sea level height oscillations. The latter are provided by shore-based tide gauges and by offshore buoys with bottom pressure sensors (tsunameters, e.g. DART buoys). First alert messages are issued from seismic information. However, assessing the tsunamigenic potential of an earthquake is not easy, so sea level measurements are needed to confirm that a tsunami was generated and to reduce the number of false alarms. Tsunami propagation models are used to forecast the time and amplitude of the wave on arrival at different coastal points, and can also be validated with sea level observations. Adequate





communications infrastructure allows issuing correct and timely warnings to local emergency management officials, who can decide to activate their emergency protocols to evacuate low-lying coastal areas in advance of the initial tsunami wave.

When the tsunami of 2004 hit the Indian Ocean causing one of the most devastating disasters of our recent history, only the Pacific Ocean had a tsunami warning system in place. Considering how many lives could have been saved, IOC/UNESCO established several intergovernmental working groups for implementation of regional and national tsunami

warning systems in other basins, such as the North-Eastern Atlantic, the Mediterranean and Connected Seas (NEAMTWS) Tsunami Warning System (UNESCO/IOC, 2012a). In 2005, most of the tide gauges in NEAMTWS region were not suitable for tsunami warning. Requirements for this application are less restrictive in terms of accuracy or datum stability than for long-term sea level trends estimates, and consist mainly of improving timeliness and lowering sampling intervals to 1 min or less for adequate measurement of tsunami wave amplitude and arrival time. A basin-wide distribution of stations is needed, with

more stations in those areas closer to tsunamigenic sources, as is the case for a significant part of the Mediterranean coast.

Following new NEAMTWS requirements for sea level data exchange, many stations have been upgraded in the M/BS region since 2005, and provide today higher-frequency sea level data in real time to regional and national tsunami warning systems. In addition, the last Global Sea Level Observing System (GLOSS) Implementation Plan (UNESCO/IOC, 2012b) suggests that GLOSS Core stations can be configured to support storm surge and tsunami warning systems. This approach has

ensured the multi-purpose character of some of the stations, good for network sustainability, but has raised new challenges on standard quality control and data processing techniques that need to be adapted and the development of automatic tools for tsunami detection (Holgate et al., 2008; Beltrami et al., 2011; Pérez Gómez et al., 2013; UNESCO/IOC, 2020). These stations are handled by national institutions and warning systems, although most of them also contribute to the IOC Sea Level Station Monitoring Facility (http://www.ioc-sealevelmonitoring.org/map.php) and the JRC-SLD https://webcritech.jrc.ec.europa.eu/

portals as well.

Based on data availability on these portals and the response from national institutions to this survey, at least 152 of all active stations in the M/BS are today contributing to tsunami warning systems, including those upgraded and those installed for this purpose (e.g. a total of 24 IDSL stations installed by JRC and 19 radar stations operated by the National Observatory of Athens (NOA) in the Greek coast). However, only three of these stations are located in the North of Africa coast: Melilla,

Alexandria and Saidia Marina. The French Tsunami Warning Center (CENALT) developed tools based on tsunami wave modelling to monitor the existing sea level network capacity for tsunami detection (Schindelé et al., 2008, 2015). As an example, they found that the tsunami generated by an earthquake North of Algeria in 2003 would not be confirmed by the tide gauge network in less than 70 min, when the travel time to the most affected zones was 30-40 min. The tool, that provides guidance for the implementation of additional tide gauges, demonstrated that adding four tide gauges, two in the Balearic

Islands, one in Sardinia, and one in Sicily, would reduce tsunami detection time by more than 20 min for sources along the North Algeria and Tunisia shoreline. They also recommend implementation of two tsunameters offshore at specific points in that area, to reduce the detection delay to less than 15-25 min along Tunisian coast. However, the high installation and



maintenance costs, along with the location of tsunamigenic areas so close to the coastline, has prevented the implementation of these offshore instrumentation in the M/BS area so far.

975        Since 2018, the Tsunami Last Mile project of Joint Research Centre (JRC) explored how to better warn the population during near shore or distant tsunami events. It was demonstrated twice, in Kos (Fall 2019, also presented at the AGU Fall Meeting in 2019) and in Malta (beginning of November 2021), how interconnecting sensing devices with an alerting system may warn the population promptly and allow a safe evacuation. Possibly, the exercise will be repeated in Indonesia during 2022. The two events differ because Kos was a near shore event and thus required the activation before the official alert from

service provider was issued, while Malta was a distant shore event so that the first activation was triggered by the information coming from the Tsunami Service Providers. The system is based on a network including IDSLs and seismometers as sensing devices and a long-range siren with two alerting panels to warn the population. As soon as both IDSLs confirmed the anomalous behaviour of the sea level, the alerting devices started alerting the population automatically, providing indications about the countermeasures to be taken. In the case of Kos this was the first triggering information. Specific signages deployed

in the streets provided indications about the evacuation routes and the assembly points.

        The system allows several degrees of automation, from completely automatic to manual activations only, depending on the standard procedures adopted by the authorities entitled to protect the population. This system is based on the multipurpose platform developed by the JRC and called RIO (Remote InterOperability), that is a complete software platform that allows easy integration of various instruments and analytical computations and activations. The RIO system was also

adopted in another project, a low cost GNSS-based buoy that provides sea level data from the open sea. Such a type of device could be therefore naturally integrated in the same network and would provide the alerting information much in advance compared with devices deployed on the coasts: depending on the installation site, the improvement in reaction time could be consistent.

        Traditionally, tide gauge stations have not been located at meteotsunami-prone areas, but rather at locations of interest

for other sea level processes (e.g. storm surges and sea level rise), and large ports and harbours where sea level data is of utmost importance for safety of navigation or at the well-inhabited coastal towns. Up to the beginning of the 21th century, at the meteotsunami hot spots the measurements have been done only during specific experiments, aimed purely at better understanding of meteotsunamis and other high-frequency sea level phenomena. Such experiments include, but are not limited to: (1) several field experiments with simultaneous air pressure and sea level measurements done during the years 1989-1992

(Monserrat et al., 1991; Rabinovich and Monserrat, 1996), and 1996-1998 (Monserrat et al., 1998; Šepić et al., 2009) in Ciutadella (the Balearic Islands, Spain); (2) field experiments, during which bottom pressure was measured at several locations within the endangered Adriatic bays, conducted through the years 2007-2008 (Orlić and Pasarić, 2008) and 2015 (not published); (3) field experiment with simultaneous air pressure and sea level measurements done in the year 2007 in Mazara del Vallo (Marrobbio Project Report, 2007; Zemunik et al., 2021a). Throughout the last two decades the national agencies and

local authorities have, however, recognised a meteotsunami threat and have been making a continuous effort towards operational monitoring and forecasting of meteotsunamis, including installation of permanent tide gauge stations at the most



endangered coastal locations. These include Ciutadella (Menorca Island, Spain), where a tide gauge station has been operational since 2013, and Vela Luka, Mali Lošinj and Stari Grad (the Adriatic Sea, Croatia), where permanent tide gauge stations have been installed during 2017-2021. Having operational stations at meteotsunami-prone locations is a prerequisite
for proper monitoring of destructive events, but more needs to be done for an efficient warning system to be implemented, given that a warning should be issued at least an hour before the meteotsunami occurs at an endangered location.

Vilibić et al. (2016) presented a design of a meteotsunami warning system based on simultaneous numerical modelling of synoptic, mesoscale atmospheric and barotropic sea level conditions, as well as a real-time assessment of atmospheric and ocean data measured a few tens up to a couple of hundreds of kilometres away from the most endangered spots, that is at
locations where off-shore meteotsunami generation and growth occur. Denamiel et al. (2019, 2021) further upgraded the concept suggesting that stochastic approaches are more reliable than deterministic numerical modelling – the prototype of such system, which gives estimates of the sea level exceedance probability during meteotsunamis, was tested for the Croatian coast of the Adriatic Sea (Denamiel et al., 2019). Similar modelling and data assessment strategies have been suggested and tested for the Balearic Islands area (Renault et al., 2011; Šepić et al., 2016a; Romero et al., 2019), where a continuously upgraded
meteotsunami warning system, with both deterministic and probabilistic component, has been operational since 1985 (Jansà and Ramis, 2021).

In spite of the great efforts invested in development of the meteotsunami warning systems, the results are still not satisfactory, resulting in a loss of trust in the early warning systems. The forecasts are known to be wrong, especially when it comes to estimating the strength and destructiveness of the event (Jansà and Ramis, 2021). The crucial problem is an intrinsic
inability of the atmospheric models – related, among else, to coarse resolutions and inadequate physical parameterization at the mesoscale – to reproduce exact properties (spatial outreach and rate of air pressure change, speed, pressure and wind spatial gradient) of the fast changing atmospheric disturbances responsible for the meteotsunami generation (Belušić et al., 2007). Slight changes of any of these properties (e.g. reducing the rate of air pressure change, translating meteotsunamigenic disturbances off their pathways for a few tens of kilometres, changing the propagation speed) may change the modelled sea
level response for several times, in particular at the most endangered areas (Vilibić et al., 2008; Orlić et al., 2010; Orfila et al., 2011; Šepić et al., 2016a; Ličer et al., 2017; Mourre et al., 2021).

Nevertheless, a significant densification of the observation network, including the sea level stations, on-shore and off-shore meteorological stations and off-shore bottom pressure recorders would help in real-time monitoring of meteotsunamigenic atmospheric disturbances and ocean conditions and in calibrating operational numerical models – and
would likely result in more reliable early warning systems.



## 4.2 Climate-related applications

### 4.2.1 Long-term variability and sea level trends

In order to estimate the long-term variability and trends of the sea level for M/BS, the PSMSL RLR dataset has been used as the primary source of monthly means (see Section 3.1). The analysis is made on a geographical extraction on M/BS of the

PSMSL relative sea level trends product (https://www.psmsl.org/products/trends/). This product fits a model composed of linear trend, seasonal component and noise on data which have at least 70% of their annual mean for the given time span. In this section four different time spans were considered all ending in 2019, the shortest one is 30 years - which is the minimum time span of the online PSMSL product and roughly corresponds to the availability of satellite altimetry – while the longest considered time span is 100 year. The number of stations per country spanning the last 30, 50, 70 and 100 years is provided in

Table 1.

Table 1: Number of stations per country available for different time spans 30, 50, 70 and 100 years in the PSMSL sea level trend online product.

| Time span period | Croatia | Italy | Spain | Greece | France | Georgia | Russia | Ukraine | Egypt | Malta | Total |
|---|---|---|---|---|---|---|---|---|---|---|---|
| 100 yr : 1920-2019 | 1 | 2 | 0 | 0 | 1 | 1 | 1 | 1 | 0 | 0 | **7** |
| 70 yr : 1950-2019 | 5 | 2 | 3 | 0 | 1 | 1 | 1 | 0 | 1 | 0 | **14** |
| 50 yr : 1970-2019 | 5 | 1 | 3 | 8 | 1 | 1 | 1 | 0 | 1 | 0 | **21** |
| 30 yr : 1990 - 2019 | 5 | 1 | 7 | 9 | 2 | 1 | 1 | 0 | 0 | 1 | **27** |

As coming from Table 1, the M/BS has few centennial tide gauge records. If we consider the records spanning the last hundred years 7 stations are available at PSMSL: 4 in the Mediterranean (Trieste and Venice in Italy, Bakar in Croatia, Marseille in France) and 3 in the Black Sea (Poti in Georgia, Tuapse in Russia and Sevastopol in Ukraine). Some stations that appear in the PSMSL database for the 1920-2019 period are not present in the shorter time spans period because they stopped with work in the second half of the twentieth century. This is the case for Sevastopol in Ukraine that had 97% of its annual

record for the period 1910 to 1994 and then stopped, so it does not show up anymore in our 70, 50 and 30 time spans.

For the sea level records spanning the last 70 years (1950-2019), 14 stations are available in the PSMSL data bank (2 times more than the centennial records). Four of these new long tide gauge records are located along the coast of Croatia (2 are located in Split, 1 in Dubrovnik and 1 in Rovinj) making the Adriatic Sea the most populated in terms of historical sea level records with half of the fourteen 70 years long records. For Italy, France and the Black Sea, the 70 and 100 years long

stations remain the same (except for Sevastopol discussed above), while Spain comprises 3 long records at the entrance of the Mediterranean Sea near Gibraltar straits (Ceuta, Malaga and Tarifa). The Alexandria station in Egypt also fits to the 70 year time span, but disappears in the shorter time span because it ended in 2006 in the PSMSL database. It is noticeable that





Alexandria is present up to 2016 in the UHSLC and that 5 of the new stations in the 70 year time span have not been updated in PSMSL since 2017, highlighting the importance of updating records within the PSMSL. Although they do not appear yet in
the PSMSL RLR dataset, some long-term records can be found from external sources. This is the case for Alicante station in Spain that has been recently digitized by Marcos et al. (2021), the Venezia station in Italy that has been updated recently (Zanchettin et al., 2021) and the data from Koper in Slovenia that has been provided by the co-authors of this paper. We have decided to add these stations in our analysis (represented in blue in Fig. 9).

If the last 30 years (1990-2019) are considered, the situation is a bit better with 27 stations available. It is noticeable
that no long-term records are available for the south of the Mediterranean Sea (North African coast). These long-term records are the only means to compute robust estimation of sea level rise.





**Figure 9: 70 years long sea level records. The curves in black are for data from PSMSL, bold are the 100 year long records, and in blue the updated or new records, not available in the PSMSL. The name of the station is followed by the relative sea level trend estimates in mm/yr and in the green boxes on the right of the curve are the vertical land motion estimates.**

In Fig. 9 the vertical land movement (VLM) rates for each station is estimated from a combination of the different GNSS solutions provided by the SONEL data center (www.sonel.org). These estimates are based on the mean of the different VLM solution from the collocated GNSS stations, and they have been supported by an update on co-location between GNSS





and tide gauges in Europe. The update has been recently released by the EuroGOOS Tide Gauge Task Team and SONEL, in the framework of a contract of EuroGOOS with the European Environmental Agency, to increase *in situ* sea level data provision to Copernicus Marine Service (https://insitu.copernicus.eu/library/reports/coins-coastal-sea level-stations). It can be
seen that all long-term records in the M/BS show relative sea level rise with values ranging from 0.68 mm/yr in Split RT Marjana in the Adriatic Sea to 7.69 mm/yr in Poti in the Black Sea. The high sea level rise at this last station suggests that it is probably prone to significant VLM (Avşar et al., 2017). One can see also the high correlation of the sea level variation in the Adriatic Sea.

### 4.2.2 Evolution of extreme sea levels

Long-term changes, at interannual and longer time scales, in extreme sea levels are primarily driven by changes in mean sea level (MSL, Woodworth et al., 2019). However, variations in extremes unrelated to MSL variability have also been identified in tide gauge records at hourly scale worldwide (Wahl and Chambers 2015; Marcos and Woodworth, 2017) and linked to changes in storminess. In the Mediterranean Sea, long-term multidecadal fluctuations in sea level extremes have been noticed in long tide gauge records, such as those in Trieste since the 1930s (Raicich, 2003) and other stations in the northern Adriatic
(Masina and Lamberti, 2013), and in Marseille since the early 20th century (Letetrel et al., 2010; Marcos et al., 2015). These variations are regionally coherent as they normally originate from large-scale atmospheric forcing (Calafat et al. 2014; Marcos et al., 2015; Lionello et al 2021). Given the variability at such low frequencies, the estimation of linear trends is highly dependent on the selected period. For example, Raicich (2003) identified a decrease in the frequency of extremes in Trieste tide gauge record during the period 1940-2000, with no clear trend in their intensity. In contrast, Masina and Lamberti (2013)
identified a small increase in the magnitude of sea level extremes in the northern Adriatic that was associated with an intensification of the bora wind in the 1990s. In Venice, Lionello et al (2021) reported that the frequency of flooding resulting from storm surges has increased since the mid-20th century, although they linked this effect to relative sea level rise rather than to a sustained trend in storminess. At interannual time scales changes in extreme sea levels are correlated with the North Atlantic Oscillation (Marcos et al., 2009b; Masina and Lamberti, 2013), even after the removal of the yearly MSL signal.

### 4.3 Satellite altimetry calibration

Satellite altimetry is nowadays an inevitable tool for mapping sea level changes over the ocean basins (Ablain et al., 2017). However, their measurements are biased by a number of processes (Fu and Haines, 2013) and therefore a proper calibration of these data is the prerequisite (Andersen and Cheng, 2013; Woodworth et al., 2017), in particular when approaching the coastal zone (Vignudelli et al., 2019). The tide gauge data is normally used for satellite altimetry calibration, yet the problem
is that the altimetry measurements are not reliable close to land, while large differences in mean sea level may occur at the coastal distances (e.g. in the last 4-5 km to the coast, as observed for the Senetosa calibration site at Corsica for both TOPEX/Poseidon and Jason altimetry missions, Gouzenes et al., 2020). Indeed, a careful examination should be conducted to verify nonexistence of all potential errors that could explain the increased rate of sea level rise close to the coast - like spurious



trends in the geophysical corrections, imperfect inter-mission bias estimate, decrease of valid data close to the coast and errors
in waveform retracking - before ascribing a finding to the real physical. Here, for the Sanatosa calibration site, it has been
proven that the steric sea level component is responsible for such changes (Dieng et al., 2021).

These problems are even more amplified in enclosed seas, such as the M/BS, where the impact of a basin topography
and a complex coastline is more pronounced than at the oceans. Orbit-related sea level errors are an example, as found to be
prominent in the Mediterranean (Esselborn et al., 2020). For that reason, calibration of satellite altimeters has a long history
there. This particularly applies to some selected tide gauge sites in the Mediterranean locations, like Ibiza (Martinez-Benjamin
et al., 2004; Frappart et al., 2015) or Corsica (Bonnefond et al., 2003, 2021; Cancet et al., 2013). Recently, advanced learning
methods (such as deep learning networks) have been used to improve calibrations of altimeter data (Yang et al., 2021).

### 4.4. Definition of vertical frames of reference

The precise information on vertical references, which strongly relies on tide gauge measurements, is also a major issue, for
both ocean and land charts. Vertical reference surfaces can be categorized under three general headings:

- tidal datum, also called chart datum which should, according to the IHO recommendations, correspond to the lowest
  astronomical sea level (LAT) or an equivalent reference level considered by the hydrographic services as being as
  close as possible to the LAT;
- vertical reference datum which is a surface of zero elevation to which heights of various points at land are referred,
being a base for defining height systems (e.g. European Vertical Reference System, EVRF2019)
- ellipsoidal reference datum, which allows to define the ellipsoidal height important for satellite altimetry
  measurements and GNSS receivers at tide gauges (e.g. Adebisi et al., 2021), such as Global Reference System 1980
  (GRS80) or World Geodetic System 1984 (WGS84).

At tide gauges, the referencing of one with respect to another is essential to allow tidal observations to serve all
possible applications (e.g. the analysis of the tidal observations allows to define the characteristics of the tide at the tide gauges
(LAT, MSL, HAT...) and consequently to determine the tidal/chart datum.

Traditionally, bathymetric data has been collected and stored relative to a tidal datum and topographic data relative
to a geodetic datum. Close to a tide gauge, bathymetric data can be referenced to the chart datum by subtracting the observations
of the tide gauge directly or associated with models. One of the most significant challenges in traditional hydrography is
establishing the relationship between the instantaneous water surface and chart datum away from water level gauge locations.
In this case, to obtain the chart depths, the vertical positions of the bottom are referred to the so-called Vertical Reference
Surface for Hydrography (VRSH), which, following the recommendations of the IHO, is identified with the development of a
3D separation model between Chart Datum (LAT) and the geodetic datum (geoid and/or respect to the ellipsoid).

In modern times, the hydrographic surveying community is using high-accuracy Global Navigation Satellite System
(GNSS) positioning techniques for vertical positioning of survey platforms, the sea surface and the sea floor. This method of
hydrographic surveying, which is known as Ellipsoidal Referenced Surveying (Hamden and Din, 2018), provides a direct





measurement of the seafloor to the ellipsoid, as established by GNSS observations, and a translation of the reference from the ellipsoid to the geoid and/or a chart datum (VRSH).

In Spain, the orthometric and ellipsoidal heights of the LAT have been established for the Iberian Peninsula and Canary Islands domains from model-reanalysis sea level data fields, which were first validated and then adjusted to experimental data from 119 tide gauge stations. The sea level height with respect to the geoid was obtained from the modelling-reanalysis service provided by the Iberia-Biscay-Ireland Monitoring and Forecasting Center (IBI MFC), in the framework of the EU Copernicus Marine Environment Monitoring System (Sotillo et al., 2015). Away from water level gauge locations the model will be adjusted in the future by GNSS water level buoys, to establish chart datum at offshore locations.

## 5 Summary and conclusions

An overview of existing coastal sea level infrastructure in the M/BS has been presented, based on the contribution from 30 institutions/sea level scientists operating tide gauges in the region. These stations are essential to monitor and study sea level variations that pose a hazard in these basins, such as storm surges, tsunamis, meteotsunamis and sea level rise, by providing accurate sea level data at all frequency ranges along the coastline. The initiative gives an insight into the status of the *in situ*
sea level network in both basins and confirms several challenging aspects such as the diversity of national strategies, sea level technology, funding or data availability, often linked to differences on the primary and evolving objectives of these installations, and their unbalanced spatial distribution. National contacts and relevant basic metadata are provided, as a starting point for improving coordination across the region. In most countries tide gauges are operated by several agencies, usually targeting different purposes. Only 7 of these institutions are in charge of more than 15 stations, while the majority operate a
smaller number of stations, sometimes one single station.

We have identified 236 active stations covering nearly all the country's coastlines in the M/BS. Several stations in Morocco and Tunisia could not be added to this inventory, and no information was obtained from Libya. confirming the lack of information along the southern Mediterranean coast from previous initiatives.

There are still many float gauges as the ones installed since the end of the XIXth century for tides and hydrography
applications, still being the second most important application, especially along the northern and eastern Adriatic and the Greek coasts. However, tides in the M/BS are generally small, so tide predictions are not computed nor needed for port operations everywhere. Therefore, an increasing number of stations are becoming multi-purpose, a term we apply here to tide gauges upgraded to be used as well in tsunami, storm surge and other early warning systems. A significant part of the network is based on radar sensors (134, 57% of the stations) providing 1 min time sampling data, with real or near-real time data transmission.
This has been mainly driven by new requirements for tsunami warning systems since 2005, that yielded to the upgrade of existing networks in several countries (e.g. Spain, France, Italy, Türkiye) and the installation of new sensors in several areas (15 new radar sensors since 2012 along the Greek coast (NOA), 24 inexpensive acoustic sensors by JRC, in collaboration with several national operators, all around the Mediterranean). In fact, about 152 of the active stations are contributing nowadays





to regional or national tsunami warning systems in this region. In addition, lower time samplings have allowed, as for the
global network, an improved dataset for understanding and warning meteotsunami events, more frequent than tsunamis at
several spots in the M/BS. While 1 min sampling is sufficient for detection of most of these sea level oscillations, access to
even higher frequency raw data from modern sensors would be desirable in the future, for a better characterization of all
periods above 30 s. As an example, PdE in Spain has recently developed a tool for operational characterization of 2 Hz raw
data from the REDMAR network, that provides these data and derived data products including waves through PdE OpenDap
server. In addition, at least 4 more institutions participating in this survey compute wind wave parameters from tide gauge raw
data.

In terms of length of the sea level records, the numbers decrease significantly: only 10 stations have been identified
with enough valid data covering the last 100 years (7 with data in the PSMSL), while 27 would have data for the last 30 years
(the altimetry period). This reflects the limitations of the network to provide reliable sea level trends and their spatial variability
along the coastline. Apart from the smaller number of tide gauges in the past, this is perhaps the most challenging application,
as it requires a precise knowledge of the station history, data archaeology efforts and often the careful combination of data
from different technologies/locations inside a harbour. Fortunately, access to VLM information has also improved in recent
years, with up to 46 stations now collocated with a permanent GNSS in the M/BS.

Some of these stations are contributing, with different time sampling, latency and quality control to one or more of
the seven international data integrators or portals described in Section 3, where a detailed assessment of data accessibility in
the M/BS is presented. The most populated portal, PSMSL, contains data from 158 stations in the region, which means that
there is still a significant number of M/BS stations focused on local or national services and not included in international
programs. This is a problem for basin scale applications and research studies and it is often related to data policy issues: not
all tide gauges provide open and free data to users, and some claim data availability only for research applications. It must be
emphasized that most of national operators rely on their own personnel and resources, including in situ maintenance as well
as quality control, data processing and product generation, and international programs have traditionally relied on these
national efforts. This requires enough funding of the networks from the Member States, not always guaranteed, and it could
also explain the lack of access to data, or the delays in updating the time series with recent records.

On the contrary, in other cases, the same sea level time series can be found at different repositories, as well as at
national data portals, with different name, metadata, or quality control, which may be confusing for end-users. These problems
are not exclusive to the M/BS, and are partly linked to the lack of unique identifiers and adequate and standard metadata
information for tide gauges. Data integrators should collaborate between them and work more closely with original data
providers, to ensure interoperability and homogeneous and good quality datasets, according to FAIR principles. These issues
are being tackled by GLOSS and by the EuroGOOS Tide Gauge Task Team in the framework of the EuroSea project (see
EuroSea Deliverable 3.3: New tide gauge data flow strategy, https://doi.org/10.3289/eurosea_d3.3).

This work shows the evolution of sea level sensors technology through the decades, for tide gauges typically installed
on a pier in a harbour. These are relatively easy to install and maintain, and the main source of in situ sea level data along the





coast. Apart from float, acoustic, pressure and radar sensors, a novel technique based on GNSS Interferometric Reflectometry (GNSS-IR) has recently emerged and revealed its potential as the number of satellite constellations increased (Peng et al., 2021). GNSS receivers have the advantage of providing both sea level and land motion information. Some institutions in the M/BS (e.g. Spanish Geographic Institute) are already exploring this technique. In the framework of the EuroSea project and the EuroGOOS Tide Gauge Task Team activities, the UK National Oceanographic Center (NOC) has recently developed a global GNSS-IR data portal, hosted at the PSMSL (https://eurosea.eu/new/a-global-sea-level-data-portal-using-global-navigation-satellite-system-interferometric-reflectometry/).

The use of GNSS receivers for sea level is becoming a reality and it is going even further. For many applications, in situ sea level measurements offshore would be a significant improvement of the coastal sea level network. One example is tsunami warning, where detection of the wave before reaching the coast is a clear advantage. GNSS receivers on buoys (GNSS-buoys) have been used for years in Japan's tsunami warning system. Despite this inventory is not including this type of stations in the M/BS, we know that several countries and agencies are now planning their implementation by adding GNSS receivers to existing or new buoys. These data would be also valuable for calibration of coastal altimetry or, as described in Section 2, for determination of offshore chart datum.

This survey has also revealed that at least 64% of active stations in the M/BS (150) have some kind of ancillary sensor providing meteorological and/or oceanographic data. Atmospheric pressure and wind are the most traditional and frequent additional parameters, very often with time samplings of 1 min, useful for meteotsunami studies. But a number of stations in the Appendix are in fact multi-sensor platforms providing a large range of parameters, as do meteorological stations (humidity, air temperature, precipitation, etc) and even ocean data like water temperature, salinity or currents. Several agencies have already or plan to install cameras (e.g. webcams planned by SHOM, in France). Multi-sensor platforms deployment seems a reasonable approach for ensuring sustainability of the networks by expanding even more their range of applications.

In summary, the assessment of the coastal sea level monitoring capacities in the M/BS exposed several important issues that are a prerequisite for making sea level operations and science over a variety of timescales and applications in these enclosed basins comprehensive: (1) a longevity of calibrated measurements is threatened at some monitoring sites, which may lower the confidence of the sea level rise estimates in the era of climate change – this should be immediately bypassed by putting again in operations all inoperable tide gauges that have multi-decadal time series; (2) the gaps in monitoring systems or their inadequacy that exist in some coastlines (e.g. Libya, Albania) should be bridged, by upgrading the existing stations or installing new ones; (3) the quoted activities should be done through collaboration and knowledge-transfer from more experienced tide gauge networks, in particular towards North Africa countries, preferably within the umbrella of existing international programmes (IOC, MONGOOS), agencies (e.g. through extending the IDSL network of JRC) or joint projects; (4) the data should be available for research and follow the open science policies, in particular of the FAIR principles (Wilkinson et al., 2016); and (5) the cacophony of sea level data repositories should be minimized, with the clear and unique provision of data for real-time and research purposes through one-stop shop service, including harmonized quality-check procedures. We hope that the next decade of coastal sea level monitoring will be as dynamic as the last decade, in which substantial progress in



some of the quoted issues has been achieved in some tide gauge networks, thus with a potential to be spilled over the whole Mediterranean and Black seas.

**Acknowledgements**

This collaborative work has been possible thanks to MONGOOS (Mediterranean Operational Network for the Global Ocean Observing System) network, aimed toward long-term synergies in the Mediterranean Sea. It has been developed in the framework of the MONGOOS Tide Gauge Task Team, as a contribution to EuroGOOS Tide Gauge Task Team and GLOSS (Global Sea Level Observing System) main objectives and activities in the M/BS. The authors would like to express their gratitude to local and national technicians, harbour personnel and experts that have been in charge of the continuous operation

and maintenance of tide gauge networks in the region, most of the times through national funding (e.g. MASRI, Bulgaria), others with the help of international programs (IOC/UNESCO, CIESM, EU framework programmes, INTERREG programmes), often facing severe difficulties to ensure permanent installations and sustainable networks. We acknowledge as well the effort of data aggregators and portals (PSMSL, GLOSS, CMEMS INS TAC, EMODnet, IOC-SLSFM, JRC-SLD, etc) on managing harmonization and generating improved data sets for scientific research. The work is partially supported by

EuroSea (EU Horizon 2020 Research and Innovation programme, grant agreement ID 862626) and JERICO-S3 (European Commission's Horizon 2020 Research and Innovation programme grant agreements No 871153).

**Author Contributions**

BPG lead this work as chair of the MONGOOS Tide Gauge Task Team and was in charge of the overall direction and planning, based on discussions with team members (IV, JS, LT, FR, MM, CF, ML, SM, GZ) for definition of the final scope and structure

of the manuscript. Several authors took the lead in writing specific sections: BPG (Sect. 1, 2 and 5), IV (Abstract, Sect. 3 and 4.3), CF (Sect. 4.1.1 and 4.4), ML (Sect. 4.1.2), JŠ (Sect. 4.1.3), LT (Sect. 4.2.1) and MM (Sect. 4.2.2). The following authors provided relevant contribution to Sect. 2, including network description, answer to the survey and stations information for their respective institutions in: Spain (BPG, EAF, MM, JT, BC, ET, AC, VMG, MAF, JMQB, MLR, JP and JS), France (CF), Italy (SM, MP, FR), Malta (AD, AG), Slovenia (MJ), Croatia (HM, SČ, DB, IM, JŠ, IV), Montenegro (BG), Albania (KZ), Greece

(GS, DAG), Cyprus (GZ), Egypt (MS), Israel (AL), Türkiye (MS, ES, HK), Bulgaria (AP), Romania (DN), Russia (IM), Algeria (HA, AM, MAM) and those from JRC, including Lebanon IDSL station (DAG). BPG designed and launched the survey to identified contacts, prepared Figs. 2 and 3, and contributed to writing in Sect. 4.1.1, 4.1.2, 4.1.3 and 4.2.1. JŠ provided meteotsunamis description in Sect. 1 and prepared Fig. 1. Maximum hourly values for Fig. 1a were provided by MM (for those stations in GESLA dataset), MS (stations in Türkiye), GS (Greek stations) and IM and SČ (Croatian stations). IM designed

and prepared Figs. 4, 5 and 6 with detailed mapping of existing infrastructure, based on the stations list in the Appendix, and IV and JŠ prepared Figs. 7 and 8 on data availability in data portals. LT compiled monthly mean sea levels from PSMSL and





several agencies, and prepared Fig. 9. JMQB and MLR contributed to Sect. 4.1.1 and 4.4, SM and MP to Sect. 4.1.2 and DAG to Sect. 5. JŠ, SČ, DB, MJ and ML helped with contacting authors from Algeria, Russia, Montenegro and Albania. Finally, IV, JS, IM and BPG helped shape the final version of the manuscript during the internal review process. All authors have read 1280 and agreed to the submission of the manuscript for publication.

**Competing interests**

The authors declare that they have no conflict of interest.



# Appendix

**Table A1**: Stations operated by Puertos del Estado (REDMAR network) and data availability.

| ID | Station | Technology | Latitude | Longitude | Data Period | Ancillary meas. | Comments |
|---|---|---|---|---|---|---|---|
| 1 | Alcudia | Radar | 39.83456 | 3.13898 | Sep 2009 - present | Atm. pressure, wind | |
| 2 | Algeciras | Radar | 36.17700 | -5.39838 | Jul 2009 - present | Atm. pressure, wind | |
| 3 | Almería | Radar | 36.83002 | -2.47835 | Jul 2006 - present | Atm. pressure, wind, GNSS | |
| | Barcelona | Acoustic | 41.34936 | 2.16023 | Aug 1992 - Dec 2007 | | Upgraded and relocated |
| 4 | Barcelona2 | Radar | 41.34177 | 2.16570 | Jan 2008 - present | GNSS | |
| 5 | Carboneras | Radar | 36.97430 | -1.89959 | Jun 2013 - present | Atm. pressure, wind | |
| 6 | Formentera | Radar | 38.73466 | 1.41903 | Sep 2009 - present | Atm.pressure, wind | |
| 7 | Gandía | Radar | 38.99521 | -0.15139 | Sep 2007 - present | | |
| | Ibiza | Pressure | 38.91123 | 1.44984 | Jan 2003 – Dec 2009 | | Upgraded |
| 8 | Ibiza2 | Radar | 38.91123 | 1.44984 | Jan 2010 - present | Atm. pressure, wind, GNSS | |
| 9 | Mahón | Radar | 39.89304 | 4.27056 | Nov 2009 - present | Atm. pressure, wind | |
| | Málaga | Acoustic | 36.71269 | -4.41545 | Jul 1992 - Jan 2010 | | Upgraded and relocated |
| 10 | Málaga3 | Radar | 36.71184 | -4.41709 | Feb 2010 - present | Atm. pressure | |
| 11 | Mallorca | Radar | 39.56015 | 2.63748 | Sep 2009 - present | Atm. pressure, wind, GNSS | |
| 12 | Melilla | Radar | 35.29061 | -2.92853 | Oct 2007 - present | GNSS | |
| | Motril | Pressure | 36.72297 | -3.52922 | Jan 2005 - May 2007 | | Upgraded and relocated |
| 13 | Motril2 | Radar | 36.72024 | -3.52360 | Jun 2007 - present | Atm. pressure, wind | |
| 14 | Sagunto | Radar | 39.63392 | -0.20624 | Sep 2007 - present | | |
| 15 | Tarifa | Radar | 36.00646 | -5.60351 | Jul 2009 - present | Atm. pressure, GNSS | |
| 16 | Tarragona | Radar | 41.07897 | 1.21325 | May 2011 - present | Atm. pressure, wind, GNSS | |
| | Valencia | Acoustic | 39.46167 | -0.32583 | Jul 1992 – Oct 2006 | | Upgraded and relocated |
| 17 | Valencia3 | Radar | 39.44203 | -0.31128 | Nov 2006 - present | | |

**Table A2**: Tide gauge stations in the Balearic Islands operated by SOCIB.

| ID | Station | Technology | Latitude | Longitude | Data Period | Ancillary meas. | Comments |
|---|---|---|---|---|---|---|---|
| 18 | Andratx | Pressure | 39.544189 | 2.378460 | Jun 2011 - present | Atm. pressure | |
| 19 | Col. Sant Pere | Pressure | 39.737317 | 3.273358 | Apr 2016 - present | Atm. pressure | |
| 20 | Pollença | Pressure | 39.904701 | 3.088516 | Jul 2009 - present | Atm. pressure | |
| 21 | Porto Cristo | Pressure | 39.539174 | 3.335090 | Mar 2016 - present | Atm. pressure | |
| 22 | Sa Rapita | Pressure | 39.360062 | 2.953671 | May 2011 - present | Atm. pressure | |
| 23 | Sant Antoni | Radar | 38.977001 | 1.298762 | Mar 2015 - present | Atm. pressure | |






**Table A3**: Stations operated by National Geographic Institute of Spain in the Mediterranean Sea and data availability.

| ID | Station | Technology | Latitude | Longitude | Data Period | Ancillary meas. | Comments |
|---|---|---|---|---|---|---|---|
| 24 | Alborán | Radar | 35.93890 | -3.03373 | Oct 2016 - present | GNSS | |
| 25 | Alicante 1 | Radar | 38.33827 | -0.47787 | May 1928 - present | GNSS | 1999: upgraded from mechanical recorder to angle encoder. 2010: upgraded from float to radar. |
| 26 | Alicante 2 | Radar | 38.33890 | -0.48123 | Mar 1957- present | Atm. pressure, GNSS | 1996: upgraded from mechanical recorder to angle encoder. 2010: upgraded from float to radar. |
| 27 | Almería | Radar | 36.83224 | -2.48499 | Jan 1990 - present | GNSS | 2010: upgraded from float to radar. |
| 28 | Cartagena | Radar | 37.59661 | -0.97385 | Jul 2002 - present | GNSS | 2010: upgraded from float to radar. |
| 29 | Cartagena | Acoustic (IDSL) | 37.567146 | -0.978958 | Jun 2018 - present | Air temperature | |
| 30 | Ceuta | Acoustic (IDSL) | 35.895837 | -5.311531 | Sep 2017 - present | Air temperature | |
| 31 | Ciutadella | Acoustic (IDSL) | 39.987588 | 3.828154 | Oct 2017- present | Air temperature | |
| 32 | La Mola de Mahon Menorca | Acoustic (IDSL) | 39.872305 | 4.308363 | Oct 2017 - present | Air temperature | |

**Table A4**: Stations operated by Spanish Institute of Oceanography in the Mediterranean Sea and data availability.

| ID | Station | Technology | Latitude | Longitude | Data Period | Ancillary meas. | Comments |
|---|---|---|---|---|---|---|---|
| 33 | Algeciras | Radar | 36.11667 | -5.43333 | 1943 - present | | 2002: upgraded from float to radar |
| 34 | Ceuta | Radar | 35.9 | -5.31666 | 1943 - present | GNSS | 2002: upgraded from float to radar |
| 35 | Palma de Mallorca | Radar | 39.55 | 2.63333 | 1997 - present | GNSS | 2002: upgraded from float to radar |
| 36 | Tarifa | Radar | 36 | -5.6 | 1943 - present | GNSS | 2002: upgraded from float to radar |

**Table A5**: Stations operated by Spanish Hydrographic Office in the Mediterranean Sea and data availability.

| ID | Station | Technology | Latitude | Longitude | Data Period | Ancillary meas. | Comments |
|---|---|---|---|---|---|---|---|
| 37 | Castellón | Acoustic | 39.96823 | 0.01988 | Sep 2021 - present | Atm. pressure, air temperature, humidity, GNSS | |
| 38 | Rosas | Acoustic | 42.25531 | 3.17819 | Sep 2021 - present | Atm. pressure, air temperature, humidity, GNSS | |




**Table A6:** Station operated by Josep Pascual (meteorological observer, Meteolestartit/CSIC).

| ID | Station | Technology | Latitude | Longitude | Data Period | Ancillary meas. | Comments |
|---|---|---|---|---|---|---|---|
| 39 | L'Estartit | Float | 42.05338 | 3.20601 | 1990 - present | Atm. pressure and other meteorological data | |

**Table A7**: Stations operated by Shom (RONIM network) and data availability.

| ID | Station | Technology | Latitude | Longitude | Data Period | Ancillary meas. | Comments |
|---|---|---|---|---|---|---|---|
| 40 | Ajaccio | Radar | 41.9227982 | 8.7628498 | 1981 - present with discontinuities | Atm. pressure | link to DATA Shom |
| 41 | Centuri | Radar | 42.965775 | 9.349833 | 2010 - present | Atm. pressure | link to DATA Shom |
| 42 | Fos Sur Mer | Radar | 43.404935 | 4.892935 | 2006 - present with discontinuities | Atm. pressure | link to DATA Shom |
| 43 | Ile Rousse | Radar | 42.6396 | 8.93524 | 2012 - present with discontinuities | Atm. pressure | link to DATA Shom |
| 44 | La Figueirette | Radar | 43.4835 | 6.93494 | 2011 - present | Atm. pressure | link to DATA Shom |
| 45 | Marseille | Radar | 43.278814 | 5.353758 | 1849 - present with discontinuities | Atm. pressure, GNSS | link to DATA Shom |
| 46 | Monaco - Fontvieille | Radar | 43.728847 | 7.421497 | 1960 - present with discontinuities | Atm. pressure | link to DATA Shom |
| 47 | Nice | Radar | 43.695508 | 7.285257 | 1981 - present with discontinuities | Atm. pressure | link to DATA Shom |
| 48 | Port Ferreol | Radar | 43.359072 | 6.717606 | 2012 - present | Atm. pressure | link to DATA Shom |
| 49 | Port Vendres | Radar | 42.519922 | 3.107450 | 1981 – present with discontinuities | Atm. pressure | link to DATA Shom |
| 50 | Port-La-Nouvelle | Radar | 43.014705 | 3.064097 | 2009 - present with discontinuities | Atm. pressure | link to DATA Shom |
| 51 | Sete | Radar | 43.397633 | 3.699105 | 1956 - present with discontinuities | Atm. pressure, GNSS | link to DATA Shom |
| 52 | Solenzara | Radar | 41.856856 | 9.40383 | 1977 - present with discontinuities | Atm. pressure | link to DATA Shom |
| 53 | Toulon | Radar | 43.11722 | 5.91306 | 1961 - present with discontinuities | Atm. pressure, GNSS | link to DATA Shom |






**Table A8:** Stations operated by ISPRA - Italian Institute for Environmental Protection and Research (RMN - National tide gauge Network; RMLV - Venice Lagoon and North Adriatic Tide Gauge Network).

| ID | Station | Technology | Latitude | Longitude | Data Period | Ancillary meas. | Comments |
|----|---------|-----------|----------|-----------|-------------|-----------------|----------|
| 54 | Ancona | Radar | 43.624 | 13.513 | 1998 - present | Atm. pressure, air temperature, wind, relative humidity, sea temperature | RMN |
| 55 | Anzio | Radar | 41.447 | 12.635 | 2011 - present | Atm. pressure, air temperature, wind, relative humidity, sea temperature | RMN |
| 56 | Bari | Radar | 41.137 | 16.861 | 1998 - present | Atm. pressure, air temperature, wind, relative humidity, sea temperature | RMN |
| 57 | Cagliari | Radar | 39.115 | 9.405 | 1998 - present | Atm. pressure, air temperature, wind, relative humidity, sea temperature | RMN |
| 58 | Canale dell'Ancora | Float | 45.524 | 12.486 | 2006 - present | | RMLV |
| 59 | Caorle | Float | 45.592 | 12.862 | 2000 - present | | RMLV |
| 60 | Carloforte | Radar | 39.148 | 8.309 | 1999 - present | Atm. pressure, air temperature, wind, relative humidity, sea temperature | RMN |
| 61 | Catania | Radar | 37.44 | 15.147 | 1999 - present | Atm. pressure, air temperature, wind, relative humidity, sea temperature | RMN |
| 62 | Cavallino Centro | Float | 45.485 | 12.551 | 1992 - present | | RMLV |
| 63 | Cavallino Darsena | Float | 45.486 | 12.586 | 2000 - present | | RMLV |
| 64 | Chioggia Diga Sud | Float | 45.229 | 12.313 | 2000 - present | Wind | RMLV |
| 65 | Chioggia Vigo | Float | 45.224 | 12.28 | 1989 - present | | RMLV |
| 66 | Civitavecchia | Radar | 42.244 | 11.554 | 1998 - present | Atm. pressure, air temperature, wind, relative humidity, sea temperature | RMN |
| 67 | Crotone | Radar | 39.023 | 17.22 | 1999 - present | Atm. pressure, air temperature, wind, relative humidity, sea temperature | RMN |
| 68 | Faro Rocchetta | Float | 45.339 | 12.311 | 1989 - present | | RMLV |
| 69 | Gaeta | Radar | 41.21 | 13.59 | 2010 - present | Atm. pressure, air temperature, wind, relative humidity, sea temperature | RMN |
| 70 | Genova | Radar | 44.41 | 8.925 | 1998 - present | Atm. pressure, air temperature, wind, | RMN |



| | | | | | | | |
|---|---|---|---|---|---|---|---|
| | | | | | relative humidity, sea temperature | |
| 71 | Ginostra | Pressure | 38.785 | 15.191 | 2010 - present | Atm. pressure, air temperature, relative humidity, sea temperature | RMN |
| 72 | Grado | Float | 45.683 | 13.383 | 1991 - present | | RMLV |
| 73 | Grassabò | Float | 45.521 | 12.53 | 1989 - present | Wind, Rain | RMLV |
| 74 | Imperia | Radar | 43.878 | 8.019 | 1998 - present | Atm. pressure, air temperature, wind, relative humidity, sea temperature | RMN |
| 75 | La Spezia | Radar | 44.097 | 9.858 | 2007 - present | Atm. pressure, air temperature, wind, relative humidity, sea temperature | RMN |
| 76 | Lampedusa | Radar | 35.5 | 12.604 | 1999 - present | Atm. pressure, air temperature, wind, relative humidity, sea temperature | RMN |
| 77 | Livorno | Radar | 43.546 | 10.299 | 1998 - present | Atm. pressure, air temperature, wind, relative humidity, sea temperature | RMN |
| 78 | Malamocco Diga Nord | Float | 45.334 | 12.342 | 2000 - present | Wind | RMLV |
| 79 | Marghera | Float | 45.474 | 12.239 | 1989 - present | | RMLV |
| 80 | Marina di Campo | Radar | 42.743 | 10.238 | 2011 - present | Atm. pressure, air temperature, wind, relative humidity, sea temperature | RMN |
| 81 | Meda Bocca Lido | Float | 45.427 | 12.415 | 1989 - present | | RMLV |
| 82 | Messina | Radar | 38.196 | 15.564 | 1999 - present | Atm. pressure, air temperature, wind, relative humidity, sea temperature | RMN |
| 83 | Murano | Float | 45.458 | 12.345 | 1995 - present | | RMLV |
| 84 | Napoli | Pressure | 40.841 | 14.269 | 1998 - present | Atm. pressure, air temperature, wind, relative humidity, sea temperature | RMN |
| 85 | Ortona | Radar | 42.356 | 14.415 | 1998 - present | Atm. pressure, air temperature, wind, relative humidity, sea temperature | RMN |
| 86 | Otranto | Radar | 40.147 | 18.497 | 1998 - present | Atm. pressure, air temperature, wind, relative humidity, sea temperature | RMN |
| 87 | Palermo | Radar | 38.121 | 13.371 | 1998 - present | Atm. pressure, air temperature, wind, relative humidity, sea temperature | RMN |
| 88 | Palinuro | Radar | 40.031 | 15.275 | 1998 - present | Atm. pressure, air temperature, wind, relative humidity, sea temperature | RMN |





| 89 | Petta de Bò | Float | 45.266 | 12.242 | 1992 - present | Wind, Rain | RMLV |
|---|---|---|---|---|---|---|---|
| 90 | Ponza | Radar | 40.867 | 12.95 | 2011 - present | Atm. pressure, air temperature, wind, relative humidity, sea temperature | RMN |
| 91 | Porto Caleri | Float | 45.095 | 12.325 | 2000 - present | | RMLV |
| 92 | Porto Empedocle | Radar | 37.286 | 13.527 | 1998 - present | Atm. pressure, air temperature, wind, relative humidity, sea temperature | RMN |
| 93 | Porto Torres | Radar | 40.842 | 8.404 | 1999 - present | Atm. pressure, air temperature, wind, relative humidity, sea temperature | RMN |
| 94 | Punta della Salute | Float | 45.431 | 12.337 | 1924 - present | | RMLV |
| 95 | Ravenna | Radar | 44.492 | 12.283 | 1998 - present | Atm. pressure, air temperature, wind, relative humidity, sea temperature | RMN |
| 96 | Reggio Calabria | Radar | 38.121 | 15.649 | 1999 - present | Atm. pressure, air temperature, wind, relative humidity, sea temperature | RMN |
| 97 | Salerno | Radar | 40.672 | 14.768 | 1999 - present | Atm. pressure, air temperature, wind, relative humidity, sea temperature | RMN |
| 98 | San Benedetto del Tronto | Radar | 42.955 | 13.89 | 2010 - present | Atm. pressure, air temperature, wind, relative humidity, sea temperature | RMN |
| 99 | San Giorgio in Alga | Float | 45.425 | 12.295 | 2000 - present | Wind, Rain | RMLV |
| 100 | San Nicolò | Radar | 45.431 | 12.383 | 1989 - present | | RMLV |
| 101 | Sant'Erasmo | Float | 45.454 | 12.386 | 1996 - present | | RMLV |
| 102 | Sciacca | Radar | 37.505 | 13.076 | 2012 - present | Atm. pressure, air temperature, relative humidity, sea temperature | RMN |
| 103 | Strombolicchio | Pressure | 38.817 | 15.252 | 2013 - present | Atm. pressure, air temperature, wind, relative humidity | RMN |
| 104 | Taranto | Radar | 40.475 | 17.225 | 1998 - present | Atm. pressure, air temperature, wind, relative humidity, sea temperature | RMN |
| 105 | Tessera | Float | 45.491 | 12.329 | 1994 - present | | RMLV |
| 106 | Tremiti | Radar | 42.119 | 15.502 | 2013 - present | Atm. pressure, air temperature, relative humidity, sea temperature | RMN |
| 107 | Treporti | Radar | 45.474 | 12.446 | 1989 - present | | RMLV |





| | | | | | | |
|---|---|---|---|---|---|---|
| **108** | Trieste | Radar | 45.649 | 13.758 | 1998 - present | Atm. pressure, air temperature, wind, relative humidity, sea temperature | RMN |
| **109** | Valle Averto | Float | 45.348 | 12.17 | 1989 - present | | RMLV |
| **110** | Venezia | Radar | 45.333 | 12.517 | 1998 - present | Atm. pressure, air temperature, wind, relative humidity, sea temperature | RMN |
| **111** | Vieste | Radar | 41.887 | 16.179 | 1998 - present | Atm. pressure, air temperature, wind, relative humidity, sea temperature | RMN |

**Table A9**: IDSL Stations operated in Italy by JRC with ISPRA.

| ID | Station | Technology | Latitude | Longitude | Data Period | Ancillary meas. | Comments |
|---|---|---|---|---|---|---|---|
| **112** | **Imperia** | Acoustic (IDSL) | 43.87834 | 8.0188 | Nov 2014 - present | Air temperature | |
| **113** | **Marina di Teulada** | Acoustic (IDSL) | 38.927535 | 8.719503 | May 2018 - present | Air temperature | |


**Table A10:** Station operated by CNR-ISMAR at Trieste, Italy, and data availability.

| ID | Station | Technology | Latitude | Longitude | Data Period | Ancillary meas. | Comments |
|---|---|---|---|---|---|---|---|
| **114** | **Trieste Molo Sartorio** | Float | 45.64725 | 13.75947 | 1875-1889 1901 - present | Atm. pressure, GNSS | Variable sampling frequency |

**Table 11:** Slovenian tide gauge station operated by the Slovenian Environment Agency, and sea level data availability.

| ID | Station | Technology | Latitude | Longitude | Data Period | Ancillary meas. | Comments |
|---|---|---|---|---|---|---|---|
| **115** | **Koper** | One radar, one float type sensor | 45.5481 | 13.7245 | 1961 - present | Atm. pressure, air temperature, wind, relative humidity, solar irradiance, sea temperature | 1961-2005: hourly 2005 - present: 10 min |

**Table A12:** Croatian tide gauge stations operated by the Andrija Mohorovičić Geophysical Institute (AMGI), Hrvatske Vode (HV), Hydrographic Institute of the Republic of Croatia (HHI), Institute of Oceanography and Fisheries (IOF), and Ruđer Bošković Institute (RBI), and data availability.

| ID | Station | Technology | Latitude | Longitude | Data Period | Ancillary meas. | Comments |
|---|---|---|---|---|---|---|---|
| **116** | **Bakar** | Float Radar | 45.3050 | 14.5400 | Dec 1929 - present | Atm. pressure | AMGI; Larger gap during World War II and a few shorter gaps |
| **117** | **Bistrina** | Radar | 42.8719 | 17.7007 | Sep 2019 - present | Atm. pressure, wind | HV |
| **118** | **Dubrovnik** | Float | 42.6579 | 18.0608 | 1955 - present | | HHI |
| **119** | **Golubinka** | Float | 44.2565 | 15.2654 | 1995 - present | | HV; Station code: 7356; No data for 2000 |
| **120** | **Mali Lošinj** | Radar | 44.53385 | 14.46358 | Jun 2021 - present | Atm. pressure, wind | IOF |
| **121** | **Ploče** | Float | 43.0481 | 17.4241 | 2003 - present | | HHI |
| **122** | **Prosinka** | Float | 43.8440 | 15.6226 | 1986 - present | | HV; Station code: 7270; Data gaps till 1994 |
| **123** | **Raslina** | Radar | 43.8073 | 15.8560 | Sep 2019 - present | Atm. pressure, wind | HV |
| **124** | **Rijeka** | Radar | 45.3241 | 14.4381 | Jun 2021 - present | | HHI; Plan to upgrade with meteorological station |
| **125** | **Rovinj** | Float | 45.0837 | 13.6291 | 1955 - present | | HHI |





| 126 | Šibenik | Radar | 43.7243 | 15.8587 | 2020 - present | Waves, surface current, sea temperature | RBI; https://hv.geolux-radars.com/sites/sibenik-svante.html |
|---|---|---|---|---|---|---|---|
| 127 | Sobra | Radar | 42.73952 | 17.62005 | May 2017 - present | | IOF |
| 128 | Split Harbor | Float | 43.5067 | 16.4386 | 1955 - present | GNSS | HHI |
| 129 | Split Marjan | Float | 43.50833 | 16.39167 | Apr 1952 - present | Atm. pressure, wind | IOF |
| 130 | Stari Grad | Radar | 43.18101 | 16.57590 | Jun 2017 - present | Atm. pressure, wind | IOF |
| 131 | Ustava ušće nizvodno | Float | 43.0073 | 17.4698 | 1990 - present | | HV; Station code: 7499; At mala Neretva River mouth; Hourly data from 1990 |
| | Vela Luka | Radar | 42.96027 | 16.70334 | May 2017- Jun 2021 | Atm. pressure, wind | IOF; Station is temporarily removed from the location; it will re-installed as soon as possible |
| 132 | Vis | Radar | 43.0619 | 16.1841 | Jun 2021 - present | Atm. pressure, wind | IOF |
| 133 | Zadar | Float | 44.1192 | 15.2304 | 1990 - present | | HHI; Larger gap during the war in Croatia |

**Table A13:** National tide gauge network of Montenegro operated by the Institute for Hydrometeorology and Seismology.

| ID | Station | Technology | Latitude | Longitude | Data Period | Ancillary meas. | Comments |
|---|---|---|---|---|---|---|---|
| 134 | Bar | Float | 42.0878 | 19.0755 | 1965 - present | | Larger GAP during 90s |
| 135 | Kotor | Float | 42.4239 | 18.7698 | 2010 - present | | |


**Table A14:** Tide gauge stations in Albania, operated by the Institute of Geoscience.

| ID | Station | Technology | Latitude | Longitude | Data Period | Ancillary meas. | Comments |
|---|---|---|---|---|---|---|---|
| 136 | Durrës | Tide pole | 41.302542 | 19.452575 | 1950 - present | | |
| 137 | Sarandë | Tide pole | 39.870519 | 20.003483 | 1950 - present | | |
| 138 | Shëngjin | Tide pole | 41.807364 | 19.586865 | 1950 - present | | |
| 139 | Vlorë | Tide pole | 40.450147 | 19.481039 | 1950 - present | | |

**Table A15:** Stations operated by the Physical Oceanography Research Group of the University of Malta.

| ID | Station | Technology | Latitude | Longitude | Data Period | Ancillary meas. | Comments |
|---|---|---|---|---|---|---|---|
| 140 | Delimara | Acoustic (IDSL) | 35.831567 | 14.554759 | Mar 2021 - present | Air temperature | Data sent in real-time to the JRC TAD Server |
| 141 | Marsaxlokk | Radar (RADAC) | 35.818083 | 14.549409 | Mar 2021 - present | Air temperature | Data sent in real-time to the JRC TAD Server |
| | Mellieha | ENDECO-type differential pressure gauge | 35.974444 | 14.351389 | May 1993 - Dec 2000 | Sea temperature | |
| 142 | Portomaso | Pressure | 35.921103 | 14.494667 | Feb 2001 - present | Atm. pressure, sea temperature | Currently offline and undergoing maintenance |



| 143 | **Senglea** | Acoustic (IDSL) | 35.889343 | 14.514121 | Jun 2021 - present | Air temperature | Data sent in real-time to the JRC TAD Server |
| 144 | **Paradise Bay** | Acoustic (IDSL) | 35.985778 | 14.331861 | 2019 - present | Air temperature | Data sent in real-time to the JRC TAD Server |

**Table A16:** Network of stations operated by the Hellenic Navy - Hydrographic Service, Greece.

| ID | Station | Technology | Latitude | Longitude | Data Period | Ancillary meas. | Comments |
|---|---|---|---|---|---|---|---|
| 145 | Alexandroupolis | Float type / digital recording | 40.84414 | 25.87827 | 1981 - present | | Upgraded to digital/GSM connection |
| 146 | Chios | Float type / analogue and digital recording | 38.37151 | 26.14119 | 1978 - present | Sea surface temperature | Upgraded to digital/GSM connection |
| 147 | Corfu | Float type / analogue recording | 39.6282 | 19.90532 | 2004 - present | | |
| 148 | Iraklio | Float type / analogue and digital recording | 35.34848 | 25.15269 | 1951 - present | Sea surface temperature | |
| 149 | Kalamata | Float type / analogue and digital recording | 37.02368 | 22.11584 | 1936 - present | Sea surface temperature, air temperature, barometric pressure, wind, humidity | Upgraded to digital/GPRS connection |
| 150 | Katakolo | Float type / analogue and digital recording | 37.64482 | 21.31968 | 1981 - present | Sea surface temperature | Upgraded to digital/GPRS connection |
| 151 | Kavala | Float type / analogue recording | 40.93464 | 24.41213 | 1933 - present | | |
| 152 | Lefkada | Float type / analogue and digital recording | 38.83454 | 20.71211 | 1979 - present | | Upgraded to digital/GSM connection |
| 153 | Leros | Float type / analogue recording | 37.12967 | 26.84799 | 1981 - present | | |
| 154 | North Chalkida | Float type / digital recording | 38.47229 | 23.59263 | 1977 - present | Sea surface temperature | |
| 155 | Patras | Float type / analogue and digital recording | 38.25937 | 21.73654 | 1975-2009 (old location), 2011 - present (new location) | Sea surface temperature | Upgraded to digital and relocated/GSM connection |
| 156 | Piraeus | Float type / analogue and | 37.93738 | 23.62671 | 1933 - present | Sea surface temperature | Upgraded to digital/GPRS |





| | | digital recording | | | | | connection |
|---|---|---|---|---|---|---|---|
| **157** | Posidonia | Float type / analogue recording | 37.95116 | 22.9601 | 1975 - present | | |
| **158** | Preveza | Float type / analogue recording | 38.95908 | 20.75663 | 1981 - present | | |
| **159** | Rhodos | Float type / digital and analogue recording | 36.44172 | 28.23644 | 1981-2009 (old location), 2010 - present (new location) | | Upgraded to digital and relocated/GSM connection |
| **160** | Salamina | Float type / digital recording | 37.97961 | 23.53664 | 1956-2004 (old location), 2009 - present (new location) | Air temperature, barometric pressure, wind, humidity, GNSS | Upgraded and relocated |
| **161** | Samos | Float type / digital recording | 37.75512 | 26.97644 | 2005 - present | | Upgraded to digital/GSM connection |
| **162** | Siros | Float type / analogue and digital recording | 37.43997 | 24.94581 | 1979 - present | | Upgraded to digital/GPRS connection |
| **163** | Skopelos | Float type / analogue recording | 39.12364 | 23.71281 | 1999 - present | | |
| **164** | Souda | Float type / digital recording | 35.48745 | 24.08248 | 1973 - present | Sea surface temperature, air temperature, barometric pressure, wind, humidity | |
| **165** | South Chalkida | Float type / digital recording | 38.46089 | 23.58946 | 1977 - present | Sea surface temperature | |
| **166** | Thessaloniki | Float type / analogue recording | 40.63254 | 22.93493 | 1933 - present | | |

**Table A17**. Network of stations operated by NOA in collaboration with JRC.

| ID | Station | Technology | Latitude | Longitude | Data Period | Ancillary meas. | Comments |
|---|---|---|---|---|---|---|---|
| **167** | Aigio | Radar | 38.25714 | 22.07685 | Oct 2013 – present | | |
| **168** | Corinth | Acoustic (IDSL) | 37.94539 | 22.93525 | Dec 2015 – present | Air temperature | |
| **169** | Greece (Lesvos Is.) | Acoustic (IDSL) | 38.97188 | 26.37055 | Nov 2019 – present | Air temperature | |
| **170** | Hrakleio | Radar | 35.3484 | 25.15254 | Oct 2013 – present | | |





| 171 | Ierapetra | Radar | 35.00374 | 25.73852 | Oct 2013 – present | |
| 172 | Itea | Radar | 38.43035 | 22.42277 | Oct 2013 – present | |
| 173 | Kalathos | Radar | 36.1139 | 28.0696 | Oct 2013 – present | |
| 174 | Kapsali | Radar | 36.1418 | 23.0037 | Jan 2012 – present | |
| 175 | Kasos | Radar | 35.4186 | 26.92184 | Oct 2013 - present | |
| 176 | Kerkyra | Radar | 39.79 | 19.91 | Oct 2013 - present | |
| 177 | Koroni | Radar | 36.7974 | 21.9602 | Jan 2012 – present | |
| 178 | Kos | Acoustic (IDSL) | 36.898362 | 27.287792 | Jul 2018 – present | Air temperature |
| 179 | Kyparissia | Radar | 37.2596 | 21.66436 | Oct 2013 – present | |
| 180 | Paleochora | Radar | 35.224 | 23.6786 | Jan 2012 – present | |
| 181 | Panormos | Acoustic (IDSL) | 38.359995 | 22.253942 | Nov 2017 – present | Air temperature |
| 182 | Plimiri | Radar | 35.9273 | 27.8575 | Oct 2013 – present | |
| 183 | Samothraki | Radar | 40.4746 | 25.46797 | Oct 2013 – present | |
| 184 | Thessaloniki | Radar | 40.63 | 22.91 | Oct 2013 - present | |
| 185 | Zakynthos | Radar | 37.78142 | 20.9052 | Oct 2013 - present | |

**Table A18:** Stations operated by the Bulgarian Coastal Sea Level Service (IO-BAS).

| ID | Station | Technology | Latitude | Longitude | Data Period | Ancillary meas. | Comments |
|----|---------|-----------|----------|-----------|-------------|-----------------|----------|
| 186 | Balchick port | Radar | 43.404162 | 28.165389 | 2009 - present | Atm. pressure, air temperature relative humidity, wind, visibility, sea temperature, salinity | |
| 187 | Burgas port | Radar | 42.483606 | 27.481925 | 1910 - present | | |
| 188 | Pmorie port | Radar | 42.551405 | 27.639129 | 2009 - present | Sea temperature, salinity | |
| 189 | Shkorpilovtci | Radar | 42.958128 | 27.900861 | 2010 - present | | |
| 190 | Varna port | Radar | 43.192504 | 27.911421 | 1919 - present | | |


**Table A19:** Stations operated by NIMRD "Grigore Antipa", Romania, and data availability.

| ID | Station | Technology | Latitude | Longitude | Data Period | Ancillary meas. | Comments |
|----|---------|-----------|----------|-----------|-------------|-----------------|----------|
| 191 | Constanta | Float and | 44.168842 | 28.658165 | 1993 - present | | The pressure sensor will be |



| ID | Station | | Latitude | Longitude | Data Period | | Comments |
|----|---------|------|----------|-----------|-------------|--|----------|
| | | pressure | 44.160805 | 28.657206 | | | replaced by a radar sensor |
| 192 | Mangalia | Pressure | 43.806973 | 28.582088 | Dec 2017 - present | | The pressure sensor will be replaced by a radar sensor |
| 193 | Sulina | Pressure | 45.162296 | 29.726634 | May 2020 - present | | |

**Table A20**: Russian and Soviet tide gauge stations operated by the All-Russian Research Institute of Hydrometeorological Information - World Data Center and data availability.

| ID | Station | Technology | Latitude | Longitude | Data Period | Ancillary meas. | Comments |
|----|---------|-----------|----------|-----------|-------------|-----------------|----------|
| | Batumi | Float | 41.7 | 41.6 | 1977–1991 | Sea temperature, salinity, wind waves | |
| | Belgorod-Dnestrovskiy | Float | 46.2 | 30.4 | 1977–1995 | Sea temperature, salinity, wind waves | |
| | Bolshoe | Float | 45.2 | 29.7 | 1977–1984 | Sea temperature, salinity, wind waves | |
| | Feodosia | Float | 45.01666 | 35.23333 | 1977 - present | Sea temperature, salinity, wind waves | |
| | Gelendzhik | Float | 44.6 | 38.1 | 1977–1992 | Sea temperature, salinity, wind waves | |
| | Heroyskoe | Float | 46.5 | 31.9 | 1985–1995 | Sea temperature, salinity, wind waves | |
| | Illichivsk | Float | 46.3 | 30.7 | 1977–1995 | Sea temperature, salinity, wind waves | |
| | Kasperovka | Float | 46.6 | 32.3 | 1977–1995 | Sea temperature, salinity, wind waves | |
| | Kherson | Float | 46.6 | 32.6 | 1977–1995 | Sea temperature, salinity, wind waves | |
| | Kulevi | Float | 42.3 | 41.7 | 1977–1979 | Sea temperature, salinity, wind waves | |
| | Nikolaev | Float | 47.0 | 32.0 | 1977–1995 | Sea temperature, salinity, wind waves | |
| | Ochakov | Float | 46.6 | 31.6 | 1977–1995 | Sea temperature, salinity, wind waves | |
| | Odessa | Float | 46.5 | 30.8 | 1977–1995 | Sea temperature, salinity, wind waves | |
| | Paromnaja Pereprava | Float | 46.3 | 30.6 | 1980–1995 | Sea temperature, salinity, wind waves | |
| | Poti | Float | 42.1 | 41.6 | 1977–1991 | Sea temperature, salinity, wind waves | |





| ID | Station | Technology | Latitude | Longitude | Data Period | Ancillary Meas. |
|---|---|---|---|---|---|---|
| | Poti (Rioni) | Float | 42.2 | 41.7 | 1977–1979 | Sea temperature, salinity, wind waves |
| | Prorva | Float | 45.5 | 29.7 | 1977–1984 | Sea temperature, salinity, wind waves |
| | Sevastopol | Float | 44.61 | 33.529 | 1977 - present | Sea temperature, salinity, wind waves |
| | Sochi | Float | 43.56510 | 39.74222 | 1977 - present | Sea temperature, salinity, wind waves |
| | Stanislav | Float | 46.6 | 32.2 | 1989–1991 | Sea temperature, salinity, wind waves |
| 194 | Tuapse | Float | 44.1 | 39.06666 | 1977 - present | Sea temperature, salinity, wind waves, GNSS |
| | Vilkovo | Float | 45.4 | 29.6 | 1977–1984 | Sea temperature, salinity, wind waves |
| | Yalta | Float | 44.48 | 34.16 | 1977 - present | Sea temperature, salinity, wind waves |


**Table A21:** Turkish Sea Level Monitoring System (TUDES) tide gauge stations operated by the General Directorate of Mapping and data availability.

| ID | Station | Technology | Latitude | Longitude | Data Period | Ancillary Meas. | Comments |
|---|---|---|---|---|---|---|---|
| 195 | Amasra | Radar | 41.74399 | 32.39033 | Jun 2001 - present | Atm. pressure, GNSS | Relocated |
| 196 | Antalya | Radar | 36.83042 | 30.60868 | Dec 1998 - present | Atm. pressure, GNSS | |
| 197 | Bodrum | Radar | 37.03218 | 27.42346 | Dec 1998 - present | Atm. pressure | |
| 198 | Bozyazı | Radar | 36.09619 | 32.94012 | Aug 2008 - present | Atm. pressure | |
| 199 | Erdek | Radar | 40.38988 | 27.84518 | Apr 1999 - present | Atm. pressure, GNSS | |
| 200 | Erdemli | Radar | 36.56372 | 34.25539 | May 2003 - present | Atm. pressure, GNSS | |
| 201 | Gazimağusa | Radar | 35.12343 | 33.95015 | Oct 2008 - present | Atm. pressure | |
| 202 | Girne | Radar | 35.34080 | 33.33406 | Oct 2008 - present | Atm. pressure, GNSS | |
| 203 | Gökçeada | Radar | 40.23171 | 25.89349 | Jan 2008 - present | Atm. pressure, GNSS | |
| 204 | İğneada | Radar | 41.88890 | 28.02352 | Jun 2002 - present | Atm. pressure, GNSS | |





| 205 | İskenderun | Radar | 36.41559 | 35.88520 | Dec 2004 - present | Atm. pressure, GNSS | Relocated |
| 206 | İstanbul | Radar | 41.15984 | 29.07413 | Feb 2011 - present | Atm. pressure, GNSS | |
| 207 | Marmara Ereğlisi | Radar | 40.96897 | 27.96215 | Jul 2004 - present | Atm. pressure, GNSS | |
| 208 | Marmaris | Radar | 36.84867 | 28.28240 | Jan 2008 - present | Atm. pressure, GNSS | Relocated |
| 209 | Menteş | Radar | 38.42961 | 26.72215 | Apr 1999 - present | Atm. pressure, GNSS | |
| 210 | Şile | Radar | 41.17637 | 29.60538 | Jan 2008 - present | Atm. pressure, GNSS | Relocated |
| 211 | Sinop | Radar | 42.02307 | 35.14946 | Jun 2005 - present | Atm. pressure, GNSS | |
| 212 | Taşucu | Radar | 36.28146 | 33.83623 | Aug 2008 - present | Atm. pressure, GNSS | |
| 213 | Trabzon | Radar | 41.00198 | 39.74455 | Jul 2002 - present | Atm. pressure, GNSS | |
| 214 | Yalova | Radar | 40.66197 | 29.27761 | Jan 2008 - present | Atm. pressure, GNSS | Relocated |

**Table A22**: JRC IDSL stations in Türkiye operated by KOERI.

| ID | Station | Technology | Latitude | Longitude | Data Period | Ancillary meas. | Comments |
|---|---|---|---|---|---|---|---|
| 215 | Bozcaada | Acoustic | 39.835741 | 26.075859 | Nov 2016 - present | Air temperature | |
| 216 | Samsun | Acoustic | 41.2949 | 36.33746 | Sep 2018 - present | Air temperature | |
| 217 | Bodrum | Acoustic | 37.031054 | 27.42477 | May 2018 - present | Air temperature | |

**Table A23:** Cyprus sea level/tide gauge stations deployed/operated by DFMR, OC-UCY, DLS, CUT, JRC, ORION

| ID | Station | Technology | Latitude | Longitude | Data Period | Ancillary meas. | Comments |
|---|---|---|---|---|---|---|---|
| 218 | Larnaca ORION | Pressure (ADCP) | 34.8836 | 33.65 | Oct 2019 - present | Waves, sea temperature, currents, susp. particles | Operational |
| | Larnaca OC-UCY | Pressure | 34.916 | 33.641 | 2010-2012 | Atm. pressure, sea temperature | Interrupted |
| 219 | Larnaca DLS | Radar | 34.928 | 33.645 | 2018 - present | Atm. pressure, sea temperature, wind, humidity | Operational |
| 220 | Limassol CUT | Radar | 34.669 | 33.042 | 2018 - present | Atm. pressure, sea temperature, wind, humidity | Operational |



| ID | Station | Technology | Latitude | Longitude | Data Period | Ancillary meas. | Comments |
|---|---|---|---|---|---|---|---|
| | Paphos DFMR | Pressure | 34.755 | 32.4087 | Sept 2001 - Jan 2013 | Atm. pressure, sea temperature | Interrupted |
| 221 | Paphos DLS | Radar | 34.7549 | 32.4075 | 2018 - present | Atm. pressure, sea temperature, wind,humidity | Operational |
| | Paralimni OC-UCY | Pressure | 35.0383 | 34.0364 | 2010-2012 | Atm. pressure, sea temperature | Interrupted |
| 222 | Paralimni DLS | Radar | 35.0384 | 34.0364 | Ma 2018 - present | Atm. pressure, sea temperature, wind, humidity | Operational |
| 223 | Pomos DLS | Radar | 35.1754 | 32.5556 | 2018 - present | Atm. pressure, sea temperature, wind, humidity | Operational |
| 224 | Zygi JRC | Radar | 34.7263 | 33.34 | 2017 - Jun 2021 | Atm.pressure, sea temperature, wind | Operational |
| | Zygi DLS | Pressure | 34.7268 | 33.34 | 2010-2012 | Atm. pressure, sea temperature | Interrupted |

**Table A24:** Tide gauge stations in Israel, operated by the National Institute of Oceanography, Israel Oceanographic and
Limnological Research.

| ID | Station | Technology | Latitude | Longitude | Data Period | Ancillary meas. | Comments |
|---|---|---|---|---|---|---|---|
| | Ashdod IDSL | Radar | 31.79633 | 34.62635 | Feb 2018 - 2020 | | |
| 225 | Ashkelon | Pressure | 31.634879 | 34.494176 | Jul 2012 - present | Atm. pressure, sea temperature, wave height and period, current, temperature, salinity, fluorescence, turbidity, oxygen | Near real time and delayed mode: https://isramar.ocean.org.il/isramar_data/TimeSeries.aspx (under Ashkelon RDI) |
| 226 | Hadera | Pressure | 32.470533 | 34.863051 | Apr 1992 - present | Atm. pressure, sea temperature, wave height and period, current, temperature, salinity, fluorescence, turbidity, oxygen | Gloss St. #80 near real time and delayed mode: https://isramar.ocean.org.il/isramar_data/TimeSeries.aspx(under Hadera RDI) |
| | Hadera IDSL | Radar | 32.47313 | 34.88207 | Feb 2018 - 2020 | | |
| 227 | Haifa IDSL | Radar | 32.822454 | 35.007042 | Feb 2018 - present | | |





**Table A25:** Stations operated by Egypt.

| ID | Station | Technology | Latitude | Longitude | Data Period | Ancillary meas. | Comments |
|----|---------|-----------|----------|-----------|-------------|-----------------|----------|
| | Abu Qir Bay | Float gauge | 31.3250 | 30.0750 | 1990-2010 | | |
| 228 | Alexandria Eastern Harbor | Radar | 31.2124 | 29.8849 | 2018 - present | | IOC Station, near real time |
| | Alexandria Western Harbour | Float gauge | 31.2001 | 29.8783 | 1979-2011 | | |
| | Burullus new harbour | Float gauge | 31.6017 | 30.9672 | 2004-2008 | | |
| | Mersa Matrouh | Float gauge | 31.3600 | 27.1833 | 2003-2006 | | |
| | Port Said | Float gauge | 31.2567 | 32.3050 | 2002-2010 | | |
| | Sidi Abdel_Raman | Float gauge | 30.9330 | 28.8360 | 2003-2010 | | |

**Table A26:** Stations operated by Algeria (National Institute of Cartography & Remote Sensing).

| ID | Station | Technology | Latitude | Longitude | Data Period | Ancillary meas. | Comments |
|----|---------|-----------|----------|-----------|-------------|-----------------|----------|
| 229 | Algiers | Analog and acoustic | 36.7681 | 3.0597 | 1985 - present 2011 - present | Atm. pressure, sea temperature, wind, air temperature, relative humidity, absolute gravity, GNSS | Upgraded to digital VPN/3G connection (Being deployed) |
| 230 | Annaba | Acoustic | 36.9000 | 7.7666 | 2016 - present | Absolute gravity, GNSS (being deployed) | Upgraded to digital VPN/3G connection (Being deployed) |
| 231 | Ghazaouet | Acoustic | 35.1000 | -1.8700 | 2015 - present | Absolute gravity, GNSS (being deployed) | Upgraded to digital VPN/3G connection (Being deployed) |
| 232 | Jijel | Analog and acoustic | 36.8209 | 5.7724 | 2004 - present 2012 - present | Absolute gravity, GNSS (being deployed) | Upgraded to digital VPN/3G connection (Being deployed) |
| 233 | Oran | Analog and acoustic | 35.7275 | -0.7012 | 2006 - present 2013 - present | Absolute gravity, GNSS (being deployed) | Upgraded to digital VPN/3G connection (Being deployed) |
| 234 | Ténès | Acoustic | 36.5233 | 1.3294 | 2014 - present | Absolute gravity, GNSS (being deployed) | Upgraded to digital VPN/3G connection (Being deployed) |





**Table A27:** Additional IDSL based stations.

| ID | Station | Technology | Latitude | Longitude | Data Period | Ancillary meas. | Comments |
|-----|---------|------------|----------|-----------|-------------|------------------|----------|
| **235** | **Batroun (Lebanon)** | Acoustic (IDSL) | 34.25848 | 35.65678 | Jun 2016 - present | Air temperature | |
| **236** | **Saïdia Marina (Morocco)** | Acoustic (IDSL) | 35.111863 | -2.292942 | Apr 2016 - present | Air temperature | |

**Supplementary material**

Response to the survey from the different networks is provided as supplementary material (Excel file).





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
