# Peer review of "Coastal Sea Level Monitoring in the Mediterranean and Black Seas"

_Ocean Science, 2021_

## Author Response (AR1)

**Reviewer 2:**

General comments:

This contribution provides a detailed review of the coastal sea level gauge network in the Mediterranean and Black Seas. On the one hand, it contains a comprehensive collection of the present-day sea-level data available in the two seas, including a detailed list of existing stations and a description of data repositories. On the other hand, a review is made of the applications that exist of such data for operational and forecasting systems of astronomical tides, storm surges, tsunamis and meteotsunamis, as well as for climate-related studies, for calibration of satellite altimetry and for the definition of height references.

The effort to collect data and metadata from all available stations and the description of the exiting repositories already makes it a useful review work for researchers and technicians that need to use sea level data, especially given the recent fast increase in the number of gauges and sensors in these seas. Moreover, they also assess the present network capabilities to fulfil important applications of sea level data and point at the most important aspects to be improved.

My minor comments, detailed below, are: their use of acronyms makes reading difficult (comment 1), a subsection on Israeli network is missing in section 2 (comment 2), including more links to data in the Tables of the Appendix would be useful (comment 3), absolute sea level rise is not included in Figure 9 and section 4.2.1 (comment 4), there are missing contents in subsections 4.2 and 4.3 (related to assessment of network capability to fulfil the targeted applications, comment 5). The use of English language is concise and correct and the article is well structured and easy to read, despite the length. Apart from two minor suggestions included in the comment 6 below, the end of this report contains a list of typos that I have encountered while reviewing the article.

All in all, this article can be of interest for many readers of Ocean Dynamics and it can be accepted with minor revisions.

**R:** We thank the positive comments about the interest of this manuscript and the nice summary of its main purpose and contribution.  We also agree with the reviewer's recommendations and have tried to answer all of them below. A list of acronyms has been provided.

Specific comments:

**Comment 1: Use of acronyms makes reading difficult**

The article uses a large amount of acronyms, which are difficult/impossible to remember so that a List of acronyms should be included. Also, some of the acronyms are used before or even without definition. Some examples are included in the list of typos at the end of the report, but it should be thoroughly revised. In this process, the authors could make an effort to decrease the number of acronyms to the minimum

(which will already be large) in order to simplify the reading of the article for the readers that are unfamiliar with the topic.

**R:** We agree that reading with so many acronyms must be difficult for someone unfamiliar with the topic. Following reviewer's recommendation we provide now a List of acronyms. However, we found reduction of the number difficult. This is partly due to many of them being referred to the different institutions, international programs, etc, which are many. The use of acronyms allow us to reduce the number of words and repetition along the text. Nevertheless, we hope the list of acronyms can help the reader.

**Comment 2: Subsection on Israeli network is missing in section 2**

In section 2.1, I would say that a subsection dedicated to sea level measurements in Israel is missing. This must be a mistake because Table A24 contains Israeli stations. Also, later on (e.g., line 627) stations in Israel are again mentioned.

R: thank you for this comment. This is indeed a mistake. IOLR has contributed providing information about stations in Israel in Table A24 and also with their response to the survey, included in the Supplementary material. We have added a brief description in Section 2.

**Comment 3: More links to data in the Tables of the Appendix would be useful**

Some of the stations in the Tables of the Appendix contain links to data. This is really useful and could be extended to many other Tables, where such links exist. This would be a good place for the authors to recommend the end users what is the best web site to download the different data, especially given the acknowledged cacophony of sea level data repositories with different degree of verification.

R:  Thank you for the recommendation. Although not always available, we have added links to national portals in these tables, when it was possible. Links to other websites are already provided in Section 3.

**Comment 4: Absolute sea level rise is not included in section 4.2.1**

If I understand it correctly, the absolute sea level rise would be computed by adding the relative sea level rise and the VLM trend, right? I would find useful that this number is added in Figure 9 and discussed in section 4.2.1. Moreover, a discussion on the error bars of VLM and the relative and absolute sea level rise values would be useful.

R: Thanks for your comment, which help us to reconsider the way we computed the error budget and clarify the Figure and the text. This is correct that the absolute sea level (ASL) trend is computed by adding the relative sea level (RSL) trend from the tide gauge with the VLM estimates from the GNSS stations. In order to complete the Figure 9 we added all these information with their uncertainties:

1. RSL trends and their uncertainties are now shown at the start of the time series inside the white box (when a robust estimation exists). When the data comes from the PSMSL datasets, the uncertainties are computed as follows (see https://www.psmsl.org/products/trends/methods.php). When we have computed the trend ourselves we used a linear regression and in that case the uncertainty is the standard error of the linear fit.

2. VLM trends and uncertainties is now shown at the end of the time series inside the light blue box (when robust estimations exist). All the estimates come from the SONEL Data Assembly Center (www.sonel.org). For any given GNSS stations, SONEL can provide up to 4 different VLM estimates depending on the analysis center (ULR, NGL, GFZ, JPL). When multiple solutions are available for a given GNSS station we have computed the weighted mean and uncertainty for this station. When multiple stations are co-located (> 15km) from a Tide Gauge with have computed the mean VLM trends of the different station. For example in the case of MARSEILLE we have 2 co-located GNSS stations (MARS, PRIE) the first one with 4 solutions and the second one with 3 solutions leading to a total of 7 solutions. The number of co-located GNSS stations and the number of solutions are now indicated after the VLM trend estimates in bracket (for example for MARSEILLE : -0.43+/-0.31 [2/7], is the mean estimates from 2 stations and 7 different solutions).

3. The absolute sea level trend is computed as the sum of the RSL and VLM and the uncertainties is simply the square root of the sum of the squared uncertainties from VLM and RSL. This ASL is now shown after the VLM estimates in the green box .

Below is the new Figure 9 and the new legend.

[Figure]

**Figure 9: 70 years long sea level records. The curves in black are for data from PSMSL, bold are the 100 years long records, and in blue the updated or new records, not available in the PSMSL. The name of the station is followed by the relative sea level trend; in the light blue boxes on the right of the curve is indicated the rate of VLM computed from different co-located GNSS stations and different solution. The number in bracket indicated the number of GNSS station / the number of solutions (i.e. [2/7] means 2 co-located GNSS and 7 different solutions) considered in the computation of the VLM estimates. The green boxes indicate the rate of absolute sea level change computed from the addition of relative sea level trend and vertical land motion. All estimates are in mm/yr.**

**Comment 5: Missing contents in subsections 4.2 and 4.3**

Sections 4.2 and 4.3, unlike section 4.1, do not exactly assess the network capability to fulfil targeted applications (the title of section 4). Subsections inside 4.1 indeed contain a technical description of how the available data and repositories of sections 2 and 3 allows performing each application of the different subsections.

On the contrary, subsection 4.1.1 at the beginning seems to focus on showing results (of sea level rise). Showing these results is highly interesting but, in order to maintain coherence, I suggest to start subsection 4.2.1 with an assessment of the capability of the available data and repositories in order to perform sea level rise analysis, and

include the results of Figure 9 at the end as an example. What I mean is that now the subsection is focussed on performing a particular analysis from the very beginning, whilst the description in the first four paragraph (up to line 1070) and Table 1 are useful (and coherent with the rest of the article) even if the results in Figure 9 were not included. In practice, this would only require of a few changes such as rephrasing the two first sentences (lines 1038-1040) to make them more general (so to present the PSMSL RLR as a tool to perform sea level rise analysis in general). Then, in line 1065, for example, a sentence should be added: "As an example of application, the sea level rise trends in the stations with 70-yr long records have been analysed (Figure 9).".

R: We agree with this point and have modified this section, taking this into account.

Section 4.2.2 only includes a (very nice) review of the results of previous studies on extreme sea levels in the MS. However, I think that the authors should also discuss if available data and repositories presented in sections 2 and 3 can be used for such analysis. Example of questions that could be answered: What are the limitations of the present M/BS network to perform such analysis? Are there gauges in the places more prone to sea level extremes? Also, no results are given in the BS. This must be simply because such studies have not been done (not because the data cannot be used for such purpose) but this should be mentioned, right?

R: We have followed this advice and have complemented the sections according to the reviewer's comments: section 4.2.2 with references to the data availability in the MS and BS. We have focused on the need of long-term records to quantify changes in extreme events. We have also pointed at the uneven spatial distribution of the tide gauge records that are suitable to do so.

In the same line, section 4.3 could extend the discussion on the usability of the present M/BS network to calibrate satellite altimetry. For example, if I understand it correctly, the lack of open sea gauges strongly limits such calibration, right? This should be added, shouldn't it? (I think it is mentioned in the conclusions, line 1225, but not here.)

R: We have also modified slightly section 4.3 in order to consider this comment from the reviewer, by adding:

*"To conclude, a very limited number of tide gauges is placed in the M/BS at locations both protruding towards the open sea and being on the track of a satellite, while there is a clear lack of open ocean buoys with a tide gauge and a GNSS receiver capable for precise positioning (Quarly et al., 2021). Indeed, such systems may minimize the errors in satellite altimetry calibrations, while state-of-the-art GNSS-based technological solution may also be deployed for improving estimates at coastal calibration sites (Bonnefond et al., 2022)."*

**Comment 6: Language corrections**

In several moments, there are single sentences (even one-line sentences) making a full paragraph, which is bizarre, right? Could you maybe integrate them into the previous

or the following paragraphs? A few examples are included in the list below (lines 339, 827-828, 847, etc.) but a thorough review should be done.

The use of "that" and "which" is sometimes wrong. A few examples are shown in the list below but a thorough review should be also performed.

R: thank you for pointing out these problems. We have thoroughly reviewed the manuscript and think both types of errors are now solved.

**Technical corrections**

R: we have corrected all minor technical corrections presented here. For those less technical we provide a more specific answer below.

Line 55 description -> a description

Line 64: being that more necessary in the era of the human-induced climate changes and the sea level rise. -> a critical need in the era of human-induced climate changes and sea level rise.

Line 71: the latter being combination -> the latter being a combination

Line 117: waves which -> waves that  or  waves, which

Line 124: gravity waves which, through -> gravity waves that, through

Line 126: situations which -> situations that

Line 154: where in some countries restrictive national data policies -> where restrictive national data policies of some countries

Line 168: Italian,French -> Italian, French

Line 171: Acknowledging all these developments, it emerged that a cohesive mapping -> A cohesive mapping

Line 172: , and even sporadically on national levels (e.g. VilibiÄ‡ et al., 2005), -> (and it has only been done sporadically on national levels, e.g. VilibiÄ‡ et al., 2005)

Line 175: of the sea level -> of sea level data

Line 194: the UNESCO -> UNESCO

Line 195: resulted with -> resulted in

Line 228: Balearic Islands -> Balearic Islands, Spain

Line 246 and 339: IDSL meaning is defined later, in line 349, right? Maybe this sentence should not be here?

Line 314: national and local level -> national and local levels

Line 335: such as: -> such as

Line 339: Is it the best place for this short sentence? I recommend to merge it within another paragraph.

Line 343: JRC and TAD meaning are still unknown at this point

Line 347 the JRC started -> the Joint Research Centre (JRC) of the European Commission started (Or it is even better to introduce it before, in section 1)

R: Done. Introduction to JRC was added to Section 1.

Line 384 and 406: I suggest to include "Float type stations:" and "Radar type type stations:" within the paragraphs (i.e., The float type stations consists of …) or just delete these words, to be coherent with the rest of subsections, which only include text within paragraphs.

Lines 429-430: was the part of Former Yugoslavia -> was part of the former Yugoslavia

Line 435: water resources -> water characteristics (?)

Line 499: preserved, -> preserved.

Line 510: aat -> at

Line 530: while in Table A22 are listed -> while Table A22 contains the

Line 575: gauges -> gauge

Line 592: observatory / observatories are accurate words? I would say this word also refers to an astronomic observatory

R: we have replaced this word by "platform", more common perhaps in oceanography.

Line 594: A subsection about the additional IDSL stations listed in Table A27 might be missing.

R: These stations belong to those described and installed by JRC (section 2.1.9), but operated jointly with receiving institutions. We have added the following to the end of Section 2.1.9:

*"JRC also operates these sensors in collaboration with other institutions in other countries, such as Greece (Table A17), Türkiye (Table A22), Cyprus (Table A23), Lebanon and Morocco (Table A27)."*

Line 602: There are many other countries with coasts on M or BS that are lacking, such as Bosnia Herzegovina, Ukraine, Georgia, Lebanon, Israel and Syria. Either a complete list should be given or the sentence should be deleted. The same applies to the Conclusion in lines 1166-1168.

R: Yes, the reviewer is right, thank you for pointing out to this. We have included these countries in the text (not Israel, as they did contribute to the manuscript).

Line 614: MSL -> mean sea level (MSL)

Line 615: mean sea level (MSL) -> MSL

Line 668: stations that have -> stations have

Line 727: the these two basins -> the two basins

Line 767: once. E.g., -> once. For example,

Line 789: in particular at hourly timescale and the raw data, not easily reachable by users -> in particular at hourly timescale, of which the raw data is not easily reachable by users (?)

Line 790: provide the access -> provide access

Line 791: purposes, and other -> purposes -, and other

Lines 827-828: Is it the best place for this short sentence? I recommend to merge it within another paragraph.

Lines 843-844: over predetermined thresholds occur -> above predetermined thresholds occur

Line 847: Is it the best place for this short sentence? I recommend to merge it within another paragraph.

Lines 913-915: Integrate the sentence into the previous paragraph?

988: , that -> , which

Line 990: a low cost GNSS-based buoy that provides sea level data from the open sea -> based on low coast GNSS buoys that provide sea level data from open sea

Line 994: Meteotsunamis are much less well known to potential readers than tsunamis. Maybe a short description of the physical phenomena could be added?

R: A short description of the phenomena is provided in the introduction, so that's why we did not add it here, to avoid repetition, and considering the length of the manuscript.

Line 1010: but more needs to be done for an efficient warning system to be implemented -> but it is not sufficient to implement an efficient warning system

Line 1011: an hour -> one hour

Lines 1015 (once) and 1033 (twice): off-shore -> offshore

Line 1020: component -> components

Line 1025: among else -> among others

Line 1030: for several times -> by an order of magnitude (?)

R: yes, this is what we meant.

Line 1032: on-shore -> onshore

Line 2069: better with -> better, with

Fig. 9: The relative sea level trend is missing in Alicante I and II.

R: These data presented several problems for the trend determination. When it was compared to Marseille trend in Marcos et al., 2021 (reference included in the manuscript), a likely datum jump was found around 1910 at Alicante I. If the trend is computed from 1911, (discarding data for this station before that year), the trend becomes 0.94 +/- 0.10 mm/yr. On the other hand, the trend for Alicante II is 0.73+/- 0.21 mm/yr, for data starting in 1957. We have added now these relative sea level trends to Figure 9.

Line 1108: the prerequisite -> a prerequisite

Lines 1112-1115: This is a dense and unclear sentence, to my opinion, and it lacks from references. First, I suggest to revise it to make it clearer and maybe split it in two sentences. Then, references should be added to sustain it. For example, I do not get what the authors mean by "the increased rate of sea level rise close to the coast". They mean that satellite altimetry gives a too large spurious sea level rise close to shore? Could they add a reference, please? Also, the list of potential errors could be referenced.

R: thank you for the comment. We have corrected this section accordingly.

*"Further, there are multiple potential errors that may affect sea level estimates, in particular close to the coast. These include spurious trends in the geophysical corrections, imperfect inter-*

*mission bias estimate, decrease of valid data close to the coast and errors in waveform retracking (Bosch et al., 2014)."*

*"To conclude, a very limited number of tide gauges is placed in the M/BS at locations both protruding towards the open sea and being on the track of a satellite, while there is a clear lack of open ocean buoys with a tide gauge and a GNSS receiver capable for precise positioning (Quarly et al., 2021). Indeed, such systems may minimize the errors in satellite altimetry calibrations, while state-of-the-art GNSS-based technological solution may also be deployed for improving estimates at coastal calibration sites (Bonnefond et al., 2022)."*

We have also provided the following references:

Bosch, W., Dettmering, D., Schwatke, C.: Multi-mission cross-calibration of satellite altimeters: Constructing a long-term data record for global and regional sea level change studies. Remote Sens., 6, 2255-2281, https://doi.org/10.3390/rs6032255, 2014.

Quartly, G.D., Chen, G., Nencioli, F., Morrow, R., and Picot, N.: An overview of requirements, procedures and current advances in the calibration/validation of radar altimeters, Remote Sens., 13, 125, https://doi.org/10.3390/rs13010125, 2021.

Bonnefond, P., Laurain, O., Exertier, P., Calzas, M., Guinle, T., and Picot, N.: Validating a new GNSS-based sea level instrument (CalNaGeo) at Senetosa Cape, Mar. Geodesy, 45, 121-150. https://doi.org/10.1080/01490419.2021.2013355, 2022.

Line 1115 to the real physical -> to real physical processes

Line 1126: datum which -> datum, which

Line 1134: the referencing of one with respect to another s essential to allow tidal observations -> the relationship between these three datum must be known to allow the observational data

Line 1136: HAT has not been defined.

Line 1136: (LAT, MSL, HAT…) -> , such as the lowest and highest levels, MSL, etc).

Line 1138: The geodetic datum should be previously defined and related with ellipsoidal datum.

Line 1155: Summary and conclusions -> Discussion and conclusions [It contains some discussion points that are not found in previous sections]

Line 1158: such as storm surges -> such as those produced by storm surges

Lines 1166-1168: Joint this sentence to the previous paragraph.

Line 1175: 2005, that -> 2005, which

Line 1187: In terms of length of the sea level records, numbers -> Focussing on long sea level records, numbers

Line 1190: Apart from the smaller number of tide gauges in the past, this is perhaps the most challenging application, as it requires -> This is perhaps the most challenging application, not only because of the smaller number of tide gauges in the past but also since it requires

Line 1227: This survey -> The present review

Lines 1227 and 1234: Paragraphs start without indentation.

Line 1231: etc) -> etc),

Line 1234: the assessment -> the present assessment

Line 1235: are a prerequisite -> could be improved

Line 1236: a longevity -> the longevity

Line 1238: operations -> operation

Line 1238: the gaps in monitoring systems or their inadequacy -> the spatial gaps in monitoring systems (or their inadequacy)

Line 1241: networks, in particular towards North Africa countries -> networks towards less experienced ones (e.g., in North African countries)

Caption Table A1: Puertos del Estado (REDMAR network) -> Puertos del Estado, Spain (REDMAR network),

Caption Table A2: Balearic Islands -> Balearic Islands, Spain,

A few other captions should be revised to add details in this line (examples are tables A6, A7, A27). The main text contains all the details but I think that the country, at least, should appear also in the Table captions.